# Genomic and phenotypic evolution of nematode-infecting microsporidia

Lina Wadi[1], Hala Tamim El Jarkass[1], Tuan D. Tran[2], Nizar Islah[1], Robert J. Luallen[2], Aaron W. Reinke[1]*

**1** Department of Molecular Genetics, University of Toronto, Toronto, Canada, **2** Department of Biology, San Diego State University, San Diego, California, United States of America

☯ These authors contributed equally to this work.
\* aaron.reinke@utoronto.ca

**Data Availability Statement:** All samples were deposited under NCBI BioProject PRJNA579982 and genome accession and Sequence Read Archive numbers are listed in S1 Table. Data from all the infection experiments is in S1 Data.

## Abstract

Microsporidia are a large phylum of intracellular parasites that can infect most types of animals. Species in the *Nematocida* genus can infect nematodes including *Caenorhabditis elegans*, which has become an important model to study mechanisms of microsporidia infection. To understand the genomic properties and evolution of nematode-infecting microsporidia, we sequenced the genomes of nine species of microsporidia, including two genera, *Enteropsectra* and *Pancytospora*, without any previously sequenced genomes. Core cellular processes, including metabolic pathways, are mostly conserved across genera of nematode-infecting microsporidia. Each species encodes unique proteins belonging to large gene families that are likely used to interact with host cells. Most strikingly, we observed one such family, NemLGF1, is present in both *Nematocida* and *Pancytospora* species, but not any other microsporidia. To understand how *Nematocida* phenotypic traits evolved, we measured the host range, tissue specificity, spore size, and polar tube length of several species in the genus. Our phylogenetic analysis shows that *Nematocida* is composed of two groups of species with distinct traits and that species with longer polar tubes infect multiple tissues. Together, our work details both genomic and trait evolution between related microsporidia species and provides a useful resource for further understanding microsporidia evolution and infection mechanisms.

## Author summary

Microsporidia are microbial parasites that can infect many animals. Nematodes have become a useful system to study microsporidia as these animals are commonly infected by microsporidia and these infections can be easily studied in a laboratory environment. To better understand how microsporidia evolve and change their properties as they infect different hosts, we sequenced the genomes of nine microsporidia species which infect nematodes. We found that metabolic pathways were mostly conserved between different clades of these species. Surprisingly, we found a family of proteins predicted to facilitate host interactions that is present in two distinct genera. We also determined the hosts and tissues that these microsporidia can infect as well as morphological properties of the

**Funding:** H.T.E.J was supported by a University of Toronto Open Fellowship. This work was supported by a Canadian Institutes of Health Research grant no. 400784 (to A.R.), an Alfred P. Sloan Research Fellowship FG2019-12040 (to A. R.), and an NSF IOS CAREER grant 2143718 (to R. L). T.T was supported by SDSU Rees-Stealy Fellowship. The funders had no role in study design, data collection and analysis, decision to publish, or preparation of the manuscript.

**Competing interests:** The authors have declared that no competing interests exist.

microsporidia. We show that these properties are quite variable in related microsporidia species. Our results allow insight into the evolution of microsporidia and provide a resource for studying microsporidia infections in nematodes.

## Introduction

Microsporidia are a large phylum of obligate intracellular parasites [1]. They were the first eukaryotic parasite to have their genome sequenced, which revealed the smallest known eukaryotic genome of 2.9 megabase [2]. In the last 20 years, genomes from ~35 microsporidian species have been sequenced, with the smallest genomes belonging to the *Encephalitozoon* which encode as few as ~1800 proteins [3]. Analysis of their genomes has shown that microsporidia are either the earliest diverging group of fungi or a sister group to fungi [4–6]. The evolution of microsporidia coincided with the loss of flagellum and the gain of a novel infection apparatus known as the polar tube [3,7]. Early-diverging microsporidian species retained a mitochondrial genome, but underwent moderate genomic loss [8,9]. Consistent with their evolution as obligate intracellular parasites, however, canonical microsporidia have undergone extensive genomic reduction, which has resulted in the loss of the mitochondrial genome and many other metabolic and regulatory proteins [10]. Additionally, microsporidian ribosomes have lost both proteins and ribosomal RNA expansion segments, resulting in the smallest known eukaryotic ribosomes [11–13]. Microsporidian proteins are also reduced, being on average ~15% shorter than their fungal orthologs [2,14]. While these larger phylogenetic changes are well studied, it is less clear how individual microsporidia genera have evolved. One interesting example is the *Enterocytozoon*, which have undergone a lineage-specific loss of many genes involved in glycolysis [15].

Microsporidia have evolved to infect a wide range of different hosts [16]. Over 1400 microsporidia species have been characterized and about half of all animal phyla as well as some protists are reported to be infected [17]. Most individual microsporidian species only infect one or two closely related hosts, but ~2% have been reported to infect more than five species. The genomic differences that account for some microsporidia having a broader host range, consistent with being a generalist, is mostly unknown, however a comparison between a pair of mosquito-infecting species showed that the generalist contains a much smaller genome than the specialist [14]. Once inside a host, most microsporidia only infect a single tissue, but ~11% can infect four or more tissues. The mechanisms that restrict microsporidia infection to specific tissues are largely unknown, but longer polar tubes correlate with infecting tissues other than the intestine, suggesting that the longer tube is needed to access these other tissues [17,18]. Interestingly, microsporidia that infect multiple host species are more likely to infect more tissues, suggesting that the selective pressures for these traits may be related [17]. Microsporidia display great diversity in their morphological (spore size, spore shape, and polar tube length) and infection properties (hosts and host tissues), even within related clades, suggesting these parasites are quite evolutionarily labile in their ability to evolve phenotypic traits [17].

Several species of microsporidia have been found to infect free-living nematodes [18–21]. The first of these to be characterized was *Nematocida parisii*, which naturally infects the model organism *Caenorhabditis elegans* [20]. This species has been used as model to understand microsporidian infection of a host, host immune responses, and identify microsporidian inhibitors [21–30]. Other species that infect the intestine, *N. ausubeli* and *N. ironsii*, have also been identified [20,21]. Another species, *N. displodere*, was found to infect other tissues of *C. elegans*, including the neurons, muscle, and epidermis [18]. Several other microsporidia

species that infect *C. elegans* or other free-living nematodes have also been identified [19]. In addition to *Nematocida*, two other genera, *Enteropsectra* and *Pancytospora*, have been reported to infect nematodes and these genera have been shown to be related to species of microsporidia that infect humans [19].

Four species of nematode-infecting microsporidia from the *Nematocida* genus have been sequenced, revealing moderate-sized genomes encoding between ~2300–2800 proteins. Many of the proteins in these genomes were found to have the potential to directly interact with their hosts, including many proteins that belong to large gene families [5,18,21]. These families were also enriched for secretion signals and transmembrane domains, suggesting that proteins in these families are used to interface with the host [10,21]. One such large gene family, NemLGF1, contains a signal for secretion and between ~160–240 proteins have been identified in each of the *N. parisii*, *N. ironsii*, and *N. ausubeli* genomes, including many lineage specific expansions [21]. This family has only been reported in these *Nematocida* species [21]. Another family, NemLGF2, is present in only *N. displodere* and contains over 230 proteins, about half of which contain a RING domain, which is thought to be involved in protein-protein interactions [19].

Here we present a comprehensive and comparative analysis of multiple genomes of nematode-infecting microsporidia. To elucidate the genomic properties of nematode-infecting microsporidia and to understand evolution within *Nematocida*, we sequenced nine additional microsporidian species, including three newly identified species. By analyzing 13 nematode-infecting microsporidian species, we show that *Nematocida* forms two groups and that *Enteropsectra* and *Pancytospora* are related to the human-infecting species *Vittaforma corneae* and together with this species form a sister group to the marine invertebrate- and human-infecting *Enterocytozoon*. We find that most conserved cellular processes are retained in *Nematocida* and that the metabolic capacity is largely similar between the genera of nematode-infecting microsporidia. We identified many novel large gene families including, surprisingly, members of NemLGF1 in *Pancytospora* species. Finally, we characterized the phenotypic properties of *Nematocida* species and describe the gain and loss of these traits throughout evolution. Our study reveals evolutionary relationships of nematode-infecting microsporidia and provides a valuable resource for the use of nematodes to study mechanisms of microsporidian infection.

## Results

### Sequencing, assembling, and phylogenetic analysis of microsporidian genomes

Previous efforts in discovering microsporidia in free-living terrestrial nematodes have revealed a diversity of species and four species from the genus *Nematocida* have been sequenced [5,18,19,21]. To further understand the genomic properties of nematode-infecting microsporidia, we sequenced nine additional species, including two genera that had not been previously sequenced. Six of these microsporidian species have been previously reported [19]. We recently discovered three novel species through sampling of wild nematode populations; *Nematocida ferruginous* from France, *Nematocida cider* from the United States, and *Nematocida botruosus* from Canada. To sequence these genomes, we removed contaminating bacteria from microsporidia-infected nematodes using antibiotics, cultured infections in the host animals, isolated spores, and extracted DNA. We sequenced a single isolate for each species, except for *N. ferruginous* for which we sequenced three strains. These strains have an average nucleotide identity to each other of greater than 99% and the different nematode-infecting microsporidia species have average nucleotide identities less than 95% (S1 Fig) [31]. Information on each microsporidia species, host, and isolation location is presented in S1 Table.

Microsporidian genomes were assembled and contaminating DNA were removed (See Methods and S2 Table). Proteins from each genome were predicted and annotated (S3 Table). These new microsporidian genome assemblies were assessed based on contig continuity and protein conservation and were demonstrated to be of high quality relative to previously assembled microsporidian genomes (Fig 1D and 1E, and S4 Table).

To determine the relative relationships between microsporidia species we performed phylogenetic analysis. Using OrthoFinder [32], we identified orthologous microsporidia proteins and generated a phylogenetic tree of our nine newly assembled microsporidia species, 35 previously sequenced microsporidia genomes, and the outgroup *Rozella allomycis* (Fig 1A and S5 Table). This tree shows that *Nematocida* genomes form two groups. The first identified *Nematocida* species, *N. parisii*, forms a group with *N. ausubeli*, *N. ironsii*, *N. major*, and

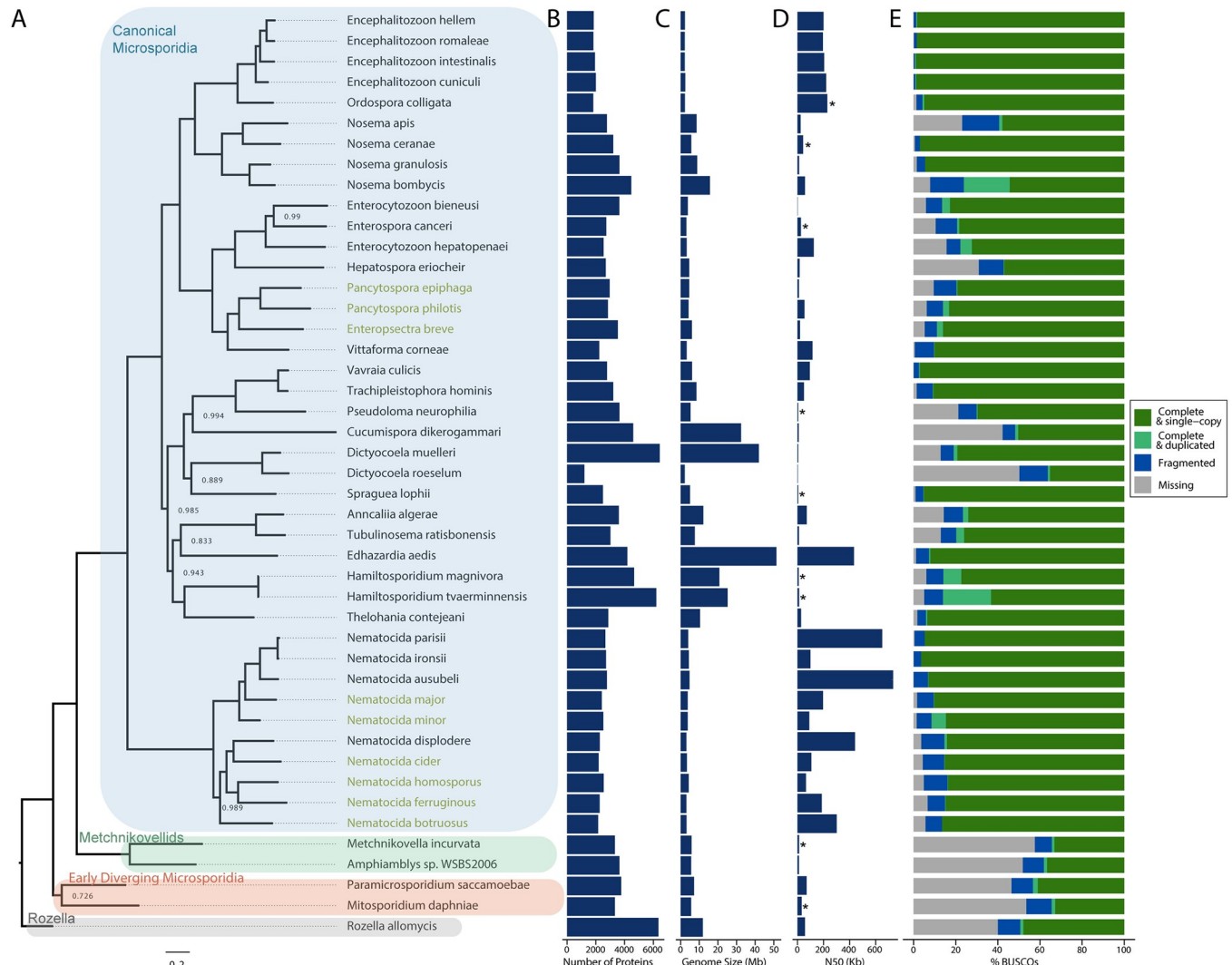

**Fig 1. Phylogeny and properties of microsporidia genomes. (A)** Phylogenetic tree of 44 microsporidia species and the outgroup *Rozella allomycis* was constructed from orthologous proteins using Orthofinder [32]. Bootstrap values less than 1.0 are indicated at each node. Scale indicates changes per site. Previously sequenced species are shown in black and newly sequenced genomes presented in this paper are shown in yellow. Species are grouped into four classes (Rozella, early diverging microsporidia, metchnikovellids, and canonical microsporidia) as previously described [3]. **(B)** Number of predicted proteins encoded by each genome. **(C)** Genome assembly size of each species. **(D)** Genome assembly N50 values. The scaffold N50 value is shown, except for those indicated by an * which are the contig N50 values. **(E)** Presence of conserved microsporidian orthologs measured as the percent BUSCOs present in each genome.

*N. minor*; we refer to these species as the "*Parisii*" group [19–21]. The second group contains the more recently identified *N. displodere* along with *N. homosporus*, *N. cider*, *N. ferruginous*, and *N. botruosus;* we refer to these species as the "*Displodere*" group [18,19]. The relative positions of the *Nematocida* species are highly supported by several phylogenetic trees (Figs 1A and S2). As described previously from 18S rDNA and β–tubulin sequences, the species *Enteropsectra breve*, *Pancytospora philotis*, and *Pancytospora epiphaga* belong to the large group of microsporidia species referred to as the Enterocytozoonida clade [17,19,33,34]. Our whole genome based tree replicates this previously described phylogeny, demonstrating that these three non-*Nematocida* species are most closely related to the human-infecting *V. corneae*, and together they form a sister group to the *Enterocytozoon* [15,35].

## Conservation and divergence of proteins in microsporidia genomes

Microsporidia have the smallest known eukaryotic genomes, but genome size and protein content can vary greatly between species [3]. In *Nematocida*, we see consistently small genome sizes that range from ~3.1 to 4.7 megabases. The two *Pancytospora* species genomes are similar to each other, with sizes of 4.2 and 4.6 megabases. In contrast, the ~6.0 megabase genome of *E. breve* is the largest of our nematode-infecting microsporidia species genomes (Fig 1C). We see similar trends in predicted number of proteins, with *N. botruosus* having the fewest proteins (2159) and *E. breve* having the most predicted proteins (3531) (Fig 1B). Although microsporidia species can have large amounts of non-coding DNA that make their genomes much larger relative to their protein content [14], we observe a strong correlation between protein content and genome size among nematode-infecting microsporidia (S3 Fig).

Microsporidia are known to have lost many conserved eukaryotic proteins, including those involved in metabolism [10]. To determine the conservation of metabolic function in nematode-infecting microsporidia species, we analyzed proteins encoded by microsporidia genomes using GO-term categories. We first determined the number of proteins from each microsporidia species that belong to a set of GO terms that cover most conserved eukaryotic cellular processes (Fig 2). This analysis revealed that many cellular processes were either lost or greatly reduced in the canonical microsporidia, which has been previously observed [8]. We then determined the proteins from each microsporidian species that belonged to eight metabolic categories (S4–S11 Figs, summarized in Fig 3). As has been previously observed, we see that the *Enterocytozoon* contains many losses in metabolic pathways, and in particular those involved in energy production [15]. As the genera *Pancytospora* and *Enteropsectra*, along with *V. cornea*, form a sister group to the *Enterocytozoon*, we sought to determine how the metabolic capacity of these groups of species compared. We observe that *Pancytospora* and *Enteropsectra* have a similar metabolic capacity compared to most other microsporidia species, including the *Nematocida* (Fig 3). However, there are several notable differences between the genera, with *Pancytospora* and *Enteropsectra* having losses in glutathione biosynthesis and the *Nematocida* having losses in phosphatidylinositol biosynthesis.

The RNAi pathway has been lost multiple times from microsporidia genomes [14,36]. *Nematocida* was previously observed to lack this pathway [3]. However, the genomes of the three species from *Pancytospora* and *Enteropsectra* each encode RNAi pathway proteins (S6 Table).

## Nematode-infecting microsporidia contain many large, expanded gene families

A striking feature of microsporidia genomes are abundant large gene families that contain paralogous family members with either signal peptide or transmembrane domains [10,21].

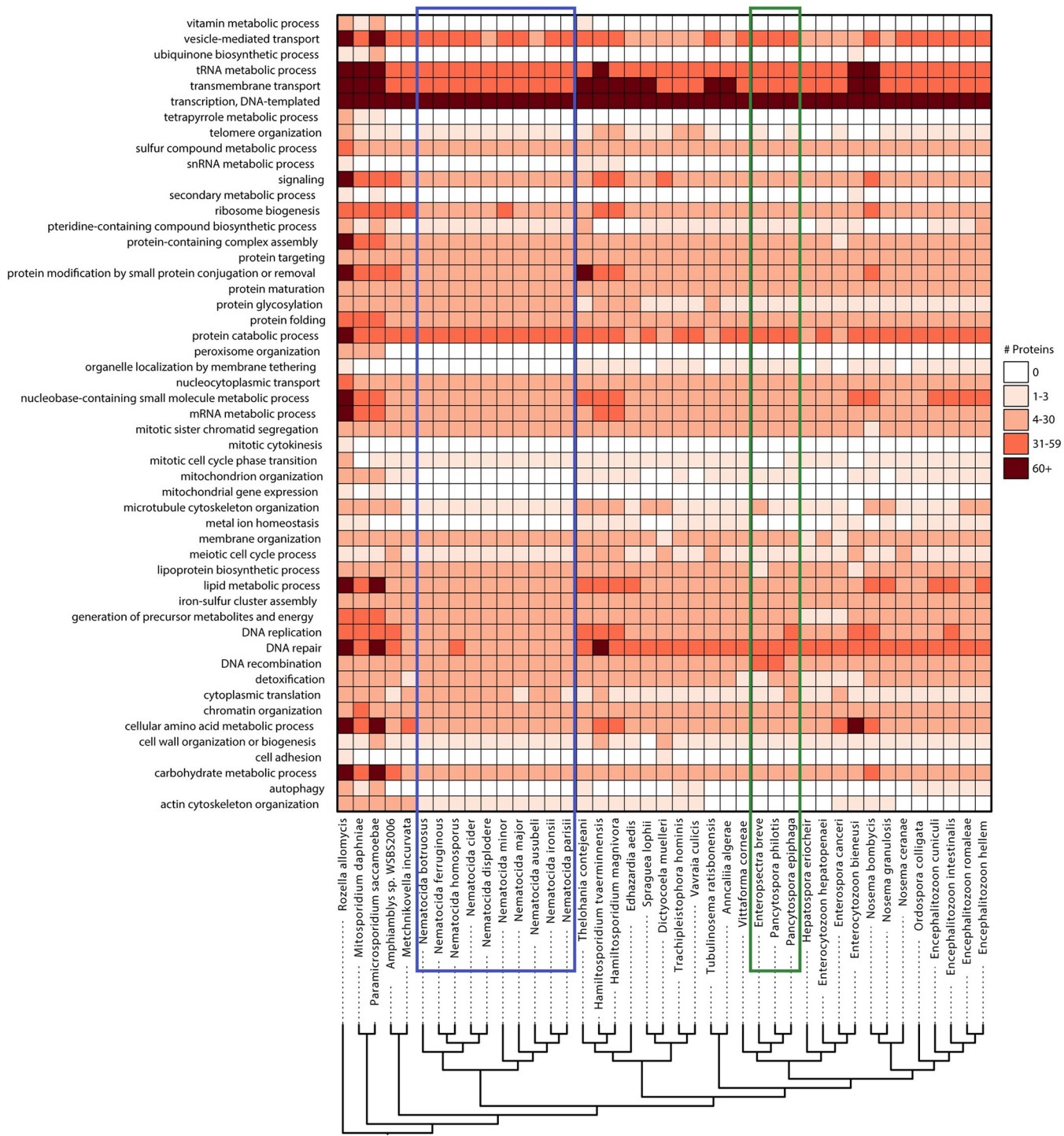

**Fig 2. Conservation of protein function across microsporidian genomes.** Membership of proteins from *R. allomycis* and 40 microsporidia species in Pombe GO-slim categories was determined. The number of proteins from each species determined to belong to each GO slim category is shown as a heatmap with GO-slim categories in rows and species in columns. Only GO-slim categories that contain at least one protein from any of these species are shown. Legend for the number of proteins in each cell is shown at the right. Phylogenetic tree, shown at bottom, was constructed using Orthofinder [32]. Several species (*Pseudoloma neurophilia*, *Dictyocoela roeselum*, *Cucumispora dikerogammari*, and *Nosema apis*) were excluded due to poorer quality genome assemblies (See Fig 1). *Nematocida* species are highlighted with a blue box. *Enteropsectra* and *Pancytospora* species are highlighted with a green box.

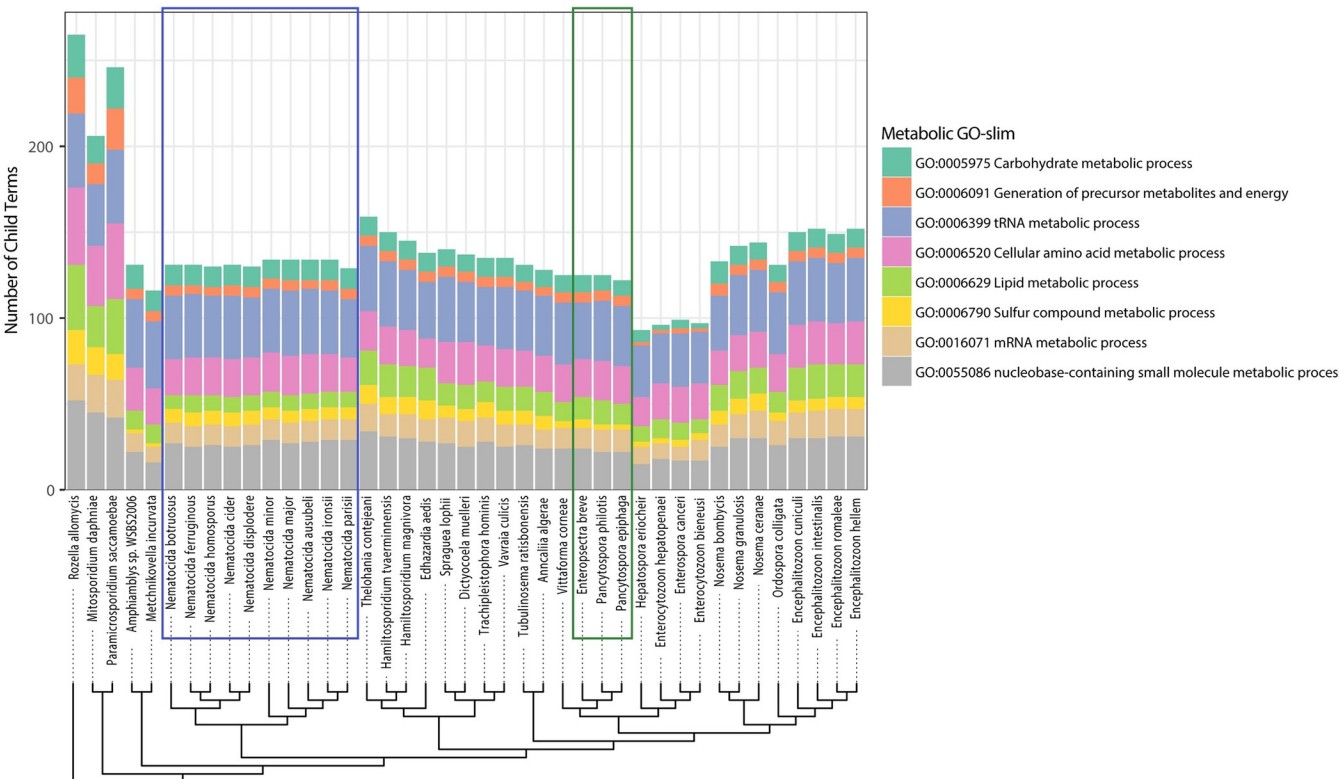

**Fig 3. Presence of metabolic enzymes in microsporidia species.** Membership of proteins from *R. allomycis* and 40 microsporidia species in eight GO-slim metabolic categories was determined. The number of proteins from each species determined to belong to each category is shown as a stacked bar graph, with the legend for each metabolic GO slim shown at the right. Phylogenetic tree, shown at bottom, was constructed using Orthofinder. Several species (*Pseudoloma neurophilia*, *Dictyocoela roeselum*, *Cucumispora dikerogammari*, and *Nosema apis*) were excluded due to poorer quality genome assemblies (See Fig 1). *Nematocida* species are highlighted with a blue box. *Enteropsectra* and *Pancytospora* species are highlighted with a green box.

Using a previously described bioinformatic approach [21], we analyzed the large gene family content of nematode-infecting microsporidia, identifying 39 families in 13 species, that were enriched (at least 50% of the proteins in the family) in either signal peptides or transmembrane domains [21] (Figs 4 and S12, and S7 Table). We observed that every species had at least one large gene family, with six species having at least five families with more than 10 members each. Some of these families have many members, with 5 families (NemLGF1, NemLGF2, NemLGF11, PanLGF1, and EbrLGF1) containing more than 100 members in at least one species. Previously, we had observed that almost all large gene families are specific to individual clades of microsporidia species [21]. Most families (30/37) in nematode-infecting microsporidia have at least 10 members in only one species. The families we identified in *Pancytospora* and *Enteropsectra* are also unique to these species and not found in *V. cornea*. Some of the large gene families, such as NemLGF11, are present in small numbers in other species, but are only greatly expanded in one of the species. To determine if any of the large gene families identified in nematode-infecting microsporidia were present in other microsporidia species, we searched other microsporidia genomes using family-specific models (S8 Table). The only family with more than 10 members present in microsporidia species that don't infect nematodes is NemLGF26, which includes the previously described InterB family that contains the Pfam domain Duf1609. This domain is also present in NemLGF26 family members [37].

One of the most abundant *Nematocida* large gene families is NemLGF1 [5,21]. This large family was previously found in the genomes of *N. parisii*, *N. ausubeli*, and *N. ironsii*, with each

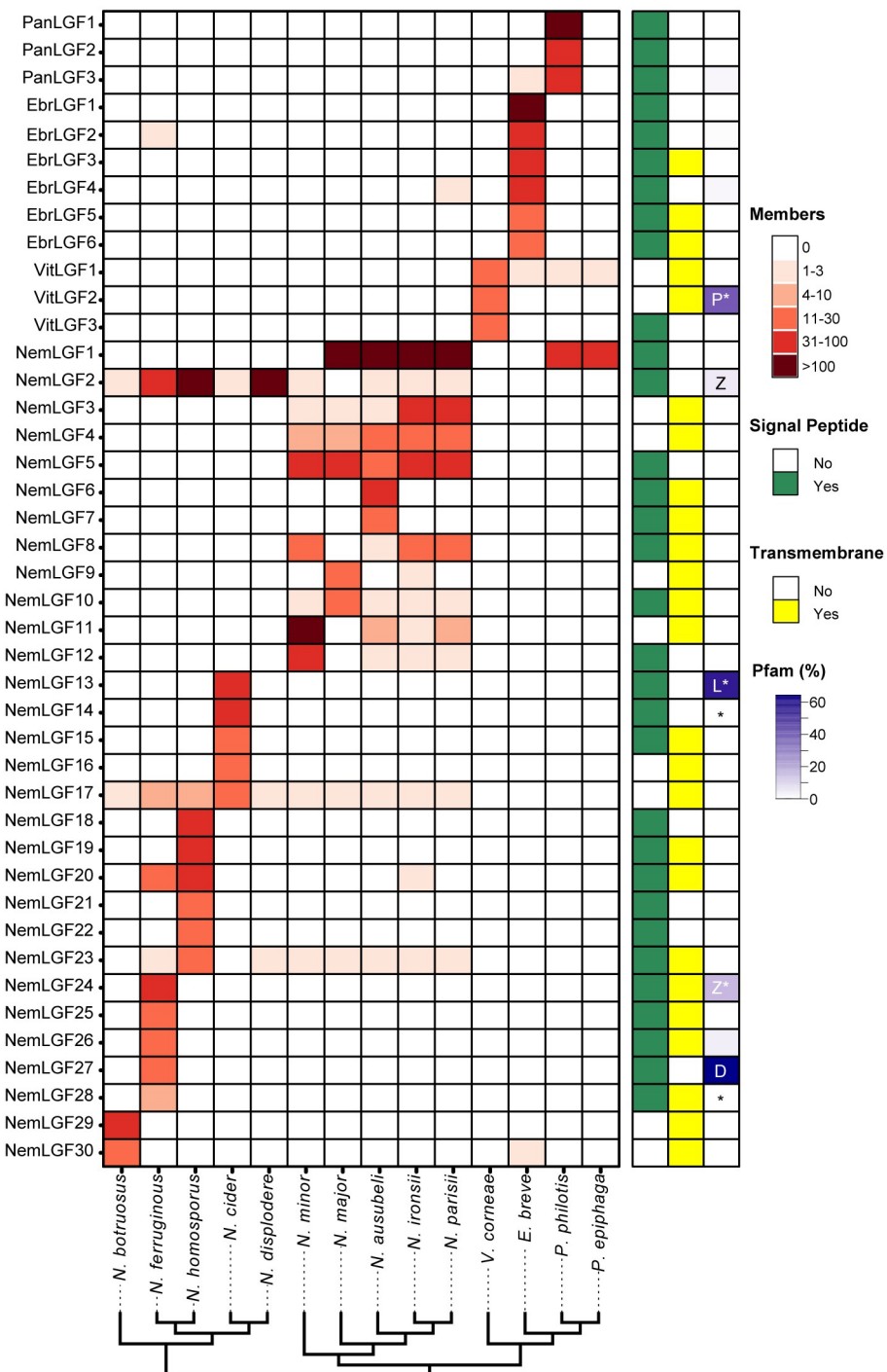

**Fig 4. Nematode-infecting microsporidia contain many diverse large gene families.** Large gene families and the number of proteins from each species that belongs to each family was determined (See methods). Large gene families were defined as at least 10 members in one species having at least 50% of the proteins in the family containing transmembrane domains or signal peptides. The number of members of each large gene family (listed in rows) in each species (listed in columns) is shown as a heatmap according to the scale at the top right. The presence of predicted signal peptides (green), transmembrane domains (yellow), or Pfam domains (purple gradient) are shown to the right. Families containing the Pfam domains peptidase (P), Zinc finger (Z), LRR(L) or DUF1609(D) are indicated. Domain types are displayed only for those families with more than one annotated Pfam domain. Families that were determined using OrthoMCL are indicated with * (See methods). Species are arranged according to the phylogenetic tree shown in Fig 1. *V. corneae* is included as an outgroup species of *Enteropsectra* and *Pancytospora*.

species containing between 160–240 proteins [21]. We also observe 105 members of this family in *N. major*. We do not observe this family in *N. minor* or any of the genomes in the *Displodere* group, suggesting that the family appeared in the *Parisii* group after the divergence of *N. minor*. Strikingly, we find that the NemLGF1 family is also present in *P. philotis* (81 members) and *P. epiphaga* (45 members). We generated a phylogenetic tree using all the members from these six species, which shows distinct expansions of NemLGF1 proteins in the *Nematocida* and *Pancytospora* species (Fig 5A).

To determine the similarity of NemLGF1 protein from *Nematocida* and *Pancytospora* species, we used AlphaFold to model the protein structures [38,39] (Fig 5B and 5C). We find that NemLGF1 proteins consist of three main domains (N-terminal, middle, and C-terminal). The N-terminal and middle domains are similar (RSDM of 1.6 Å for N-terminal domain and 1.2 Å for middle domain) between *N. parisii* member NEPG_02057 and *P. epiphaga* member PAEPH01_0103 (Fig 5D and 5E). However, the C-terminal domains form a bundle of alpha helices that are less conserved (RSDM of 18.3 Å) (Fig 5F). We also compared an *N. ausubeli* NemLGF1 member NERG_01890 to *N. parisii* NEPG_02057, and observed similar trends, although the C-terminal domains were more conserved (RSDM 4.5 Å) than with *P. epiphaga* PAEPH01_0103 (S13A–S13E Fig). These high overall structural similarities between NemLGF1 proteins exist despite the sequence identities being only 22.03% between NEPG_02057 and PAEPH01_0103 and 27.37% between NEPG_02057 and NERG_01890 [40]. To confirm that this family does not exist in any other species of microsporidia, we searched the predicted NEPG_02057 structure against a database of predicted structures and no statistically significant matches to proteins in other microsporidia species were identified [41].

To begin to understand the biochemical function of these nematode infecting large gene families, we generated structural models for the four other largest families using Alphafold (S13F–S13I Fig). None of these families have apparent structural similarity to other solved proteins, with the exception of NemLGF2, which shares similarity to Leucine Rich Repeat proteins (S13I Fig), which is a domain that has been previously observed in microsporidia proteins [21,42]. This family was previously described in *N. displodere* and we also observe large expansions of this family in *N. ferruginous* and *N. homosporous*, but not in *N. cider*. As previously shown, although we detect a member of this family in the *Parisii* group species, we only observe expansions of this family within some *Displodere* group species.

### Evolution of *Nematocida* phenotypic traits

Microsporidia often only infect one or more closely related host species, but generalists that infect many hosts have also been observed [16]. Previous work in *Nematocida* have described species with more restricted host specificity, such as *N. parisii* and *N. ausubeli*, as well as those with a broader host range, like *N. displodere* and *N. homosporus* [19,43]. To characterize the host range of the novel species we isolated, we set up experimental infections against seven different nematode hosts. As controls, we also challenged each of these hosts with *N. parisii* and *N. ausubeli*. After 4 days of infection, we fixed and stained populations of animals using direct yellow 96 (DY96) [22,44] and quantified the proportion of the population that produced microsporidia spores (Fig 6A). For *N. parisii* and *N. ausubeli*, we observed strong infection of *C. elegans* and *Caenorhabditis briggsae*, as previously reported [19]. *N. ausubeli* also infected a modest percentage of *Caenorhabditis tropicalis* and a small percentage of *Caenorhabditis nigoni*, while *N. parisii* infected a small percentage of *Caenorhabditis tropicalis*. These two species could not infect any of the other hosts. These results are consistent with what has been previously reported [19]. For *N. botruosus*, we only observed infection of its native host (*Panagrellus* sp. 2*)*, and none of the other six host species tested. In contrast, *N. cider* could

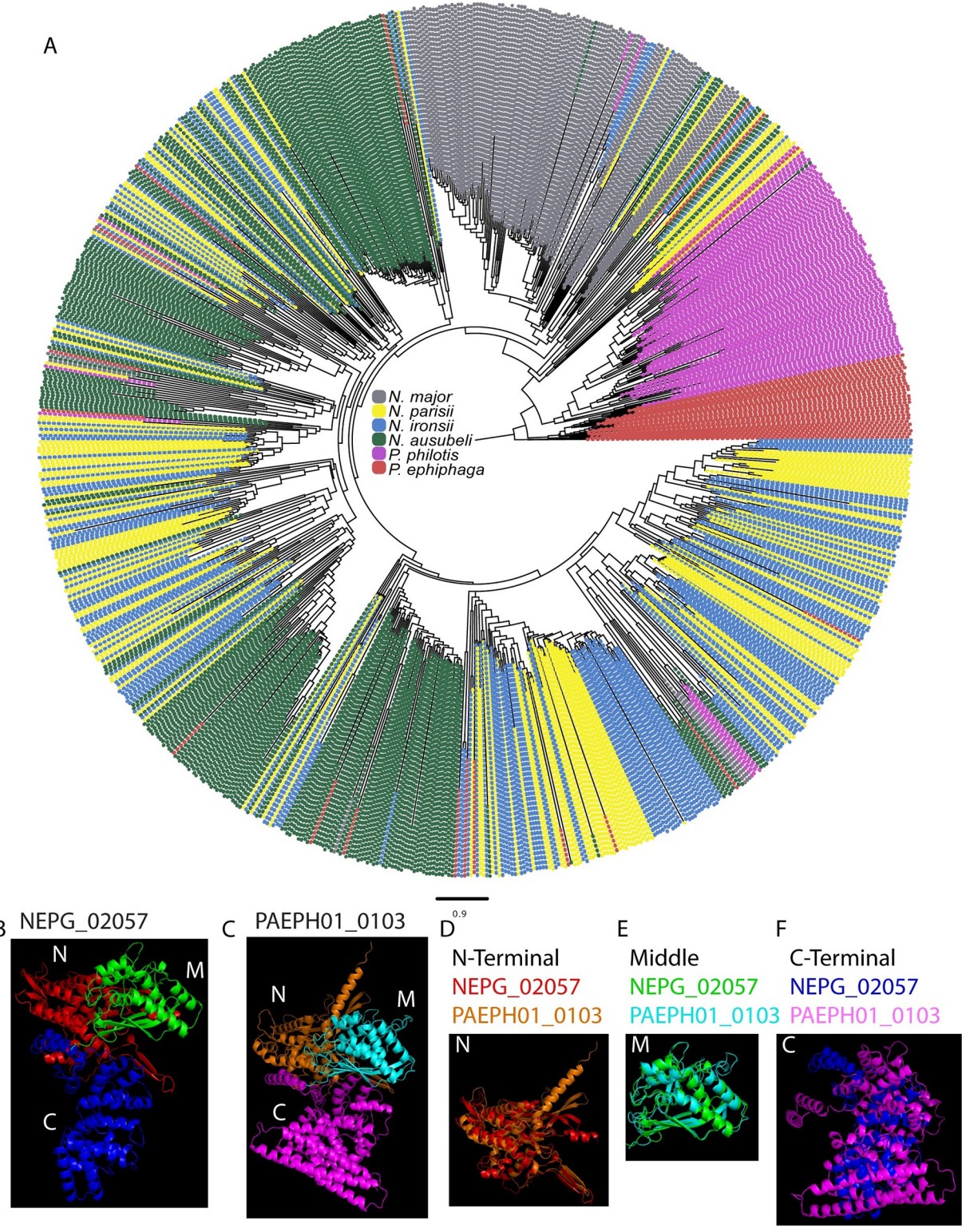

**Fig 5. The large gene family NemLGF1 has likely been horizontally transferred between genera.** (A) Phylogenetic tree of NemLGF1 family members, colored by species according to the legend in the middle of the tree. Scale indicates changes per site. (B-C) AlphaFold models of *N. parisii* NEPG_02057 (B) and *P. epiphaga* PAEPH01_0103 (C). (D-F) Aligned structures of the N-terminal (D), middle (E), and C-terminal (F) domains. N, N-terminal. M, middle. C, C-terminal.

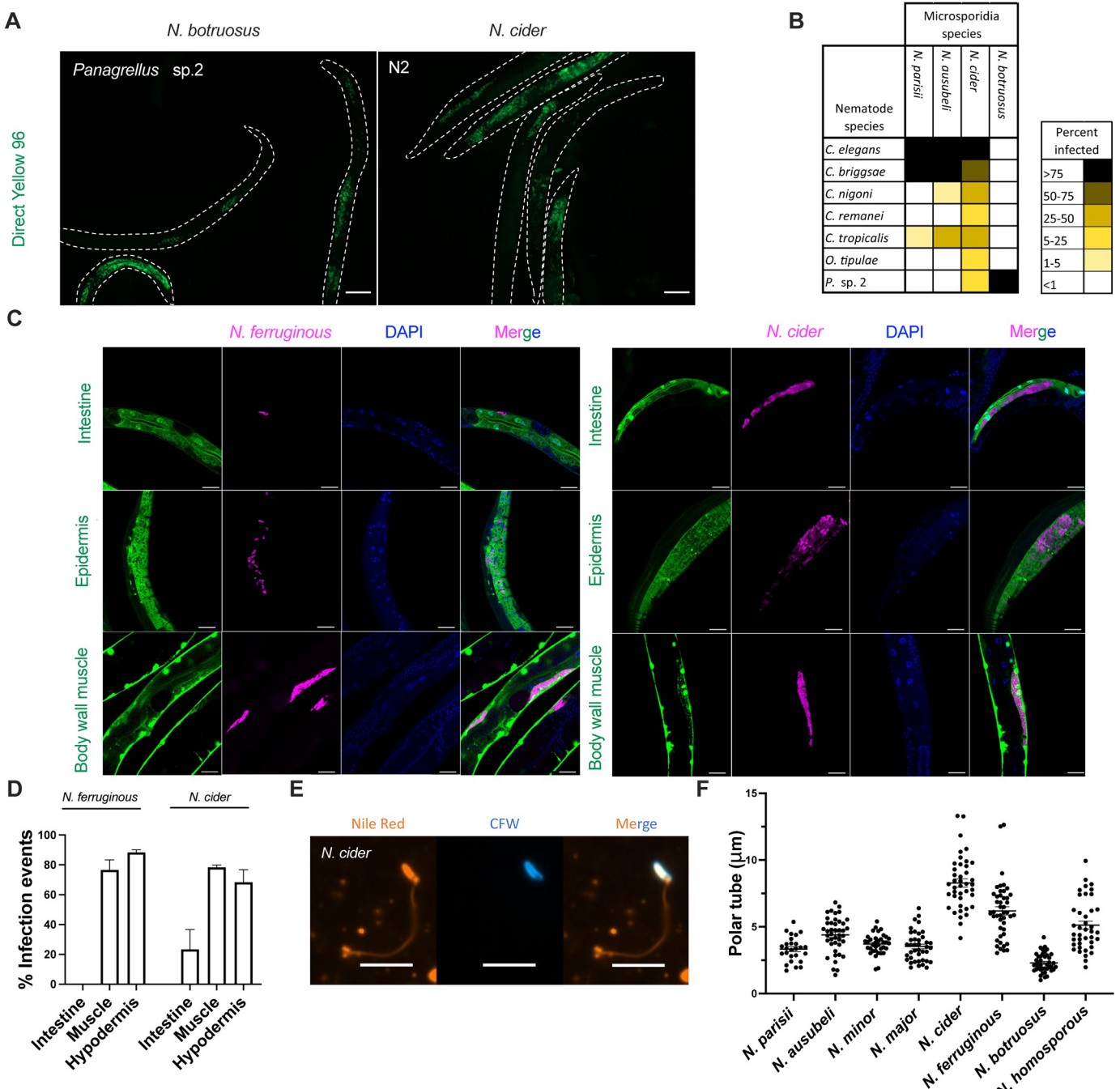

**Fig 6. Determination of host range, tissue specificity, and polar tube length of *Nematocida* species. (A-B)** Seven species of L1 stage nematodes were infected with the following species (spore concentration): *N. parisii* (3 million), *N. ausubeli* (0.8 million), *N. cider* (4 million), and *N. botruosus* (28 million). After 96 hours of incubation with spores, animals were fixed and stained with DY96. **(A)** Representative images of *C. elegans* infected with *N. cider* and *Panagrellus* sp. 2 infected with *N. botruosus*. Scale bars, 100 μm. **(B)** Percentage of each population of animals infected with each species of microsporidia. Data is displayed as a heat map with host species in rows, microsporidia species in columns, and the value of each cell being the percentage of each population that displayed newly formed microsporidia spores. Legend is displayed on the right. Data shown are the average of two biological replicates with 20–391 animals counted for each sample. **(C-D)** L1 stage *C. elegans* strains expressing GFP in either the intestine, epidermis, or body wall muscle were infected with either 1 million *N. cider* or *N. ferruginous* spores. After 72 hours animals were fixed and stained with a *Nematocida* 18S RNA FISH probe and DAPI. Note that for *N. ferruginous* infection of the GFP-intestinal strain (top), that the infection is seen to lie outside of the GFP signal indicating a lack of intestinal infection. **(C)** Representative images of infection of each species in each strain. Scale bars, 20 μm. **(D)** The percentage of the population of each strain containing microsporidia infection within the marked tissue. Data are from over 2 biological replicates and a total of 60 animals examined for each condition. Graphs show mean with SD error bars. **(E-F)** Spores of *Nematocida* species were induced to fire through repeated freeze thawing and then stained with Nile red and calcofluor white. **(E)** Representative image of a stained *N. cider* polar tube. Scale bars, 5 μm. **(F)** The length of polar tubes of each species is displayed. 26–41 polar tubes were measured for each species. Mean ± SEM represented by horizontal bars.

infect all the hosts tested to some extent (Fig 6B), with *C. elegans* being the most infected at 85.6% and *Panagrellus* sp. 2 being the least infected at 7.4%. For *N. ferruginous*, we were unable to observe more than 50% of animals infected for any nematode species, even when using a high concentration of spores. However, even at this level of infection, we observed infection of *C. elegans*, *C. briggsae*, and *C. tropicalis*. Additionally, we saw infection in *Caenorhabditis remanei* and *Oscheius tipulae*, which were species that none of the other microsporidia besides *N. cider* could infect (S14 Fig). Our data suggests that *N. botruosus* has a specialized host range, whereas *N. cider* and *N. ferruginous* appear to be generalists.

Microsporidia mostly infect only one tissue, but some microsporidia, including *N. displodere*, have been shown to infect multiple tissues [17]. During our infection experiments, we only ever observed spore formation in the intestine for *N. botruosus*, but observed infection in other tissues for *N. cider* and *N. ferruginous* (S15A–S15C Fig). Additionally, during prolonged infection with *N. cider* or *N. ferruginous*, the entire animal turns brown, suggesting that infection is occurring throughout the animal (S15D Fig). To further characterize which tissues *N. cider* and *N. ferruginous* infect, we used *C. elegans* strains with GFP specifically expressed in either the intestine, epidermis, or body wall muscle (Fig 6C). We observed *N. ferruginous* could infect both the muscle and the epidermis but we did not see productive infection in the intestine. In contrast, *N. cider* infection was seen in all three tissues, the muscle, epidermis, and intestine, although at a lower frequency in the intestine (Fig 6D).

We recently showed that the *C. elegans* protein AAIM-1 is necessary for efficient invasion by *N. parisii* [23]. This protein is secreted and likely functions in the intestinal lumen. Although *N. displodere* infects other tissues besides the intestine, feeding is necessary for infection, implying that spores need to first be in the intestinal lumen to infect tissues beyond the intestinal cells. To determine if infection by an epidermal and muscle infecting species was also dependant upon this protein, we infected wild-type and *aaim-1* mutant animals with *N. cider*. Similar to what was observed with *N. parisii* infection, *aaim-1* mutant animals exposed to a high dose of *N. cider* have a reduced prevalence of infection compared to wild-type animals (S16 Fig).

Microsporidia species have an extracellular spore form of a defined morphology which can be used to identify species [17]. *Nematocida* spores have been shown to have a variety of sizes, with the average size of species ranging from 1.3–2.38 μm long and 0.53–1.03 μm wide. Additionally, some species in the *Parisii* group have been shown to have a second, larger, class of spores [19]. To determine spore sizes for the new species we identified, we examined infected animals with stained spores (S15A–S15C Fig). This revealed that these species only have a single class of spores, with *N. botruosus* measuring 1.51 x 0.76 μm, *N. cider* measuring 2.4 x 0.67 μm, and *N. ferruginous* measuring 1.72 x 0.6 μm.

Microsporidia spores have a characteristic infection apparatus known as the polar tube [45]. The length of this tube was previously shown to correlate with tissues infected [17]. For example a species with a shorter polar tube (~ 4 μm for *N. parisii*) infects the intestine, and a species with a longer polar tube (~ 12 μm for *N. displodere*) can infect the muscles, neurons, and epidermis [18]. This correlation is hypothesized to be due to the requirement of longer polar tubes to access more distal tissues from the lumen of the intestine [17,18]. We measured polar tube lengths of several *Nematocida* species using a previously described freeze-thaw approach to trigger spore firing [18] followed by staining with Nile red [46] (Fig 6E). We observed a range of lengths throughout *Nematocida*, with the shortest being *N. botruosus* with a length of 2.3 μm and the longest being the previously measured *N. displodere* [18] (Fig 6F).

To understand how various microsporidian phenotypic traits evolve, we plotted our measured attributes onto the phylogenetic tree of *Nematocida* (Fig 7). We observe that all of the *Parisii* group species have a specialized host range and show specificity for the intestine. In

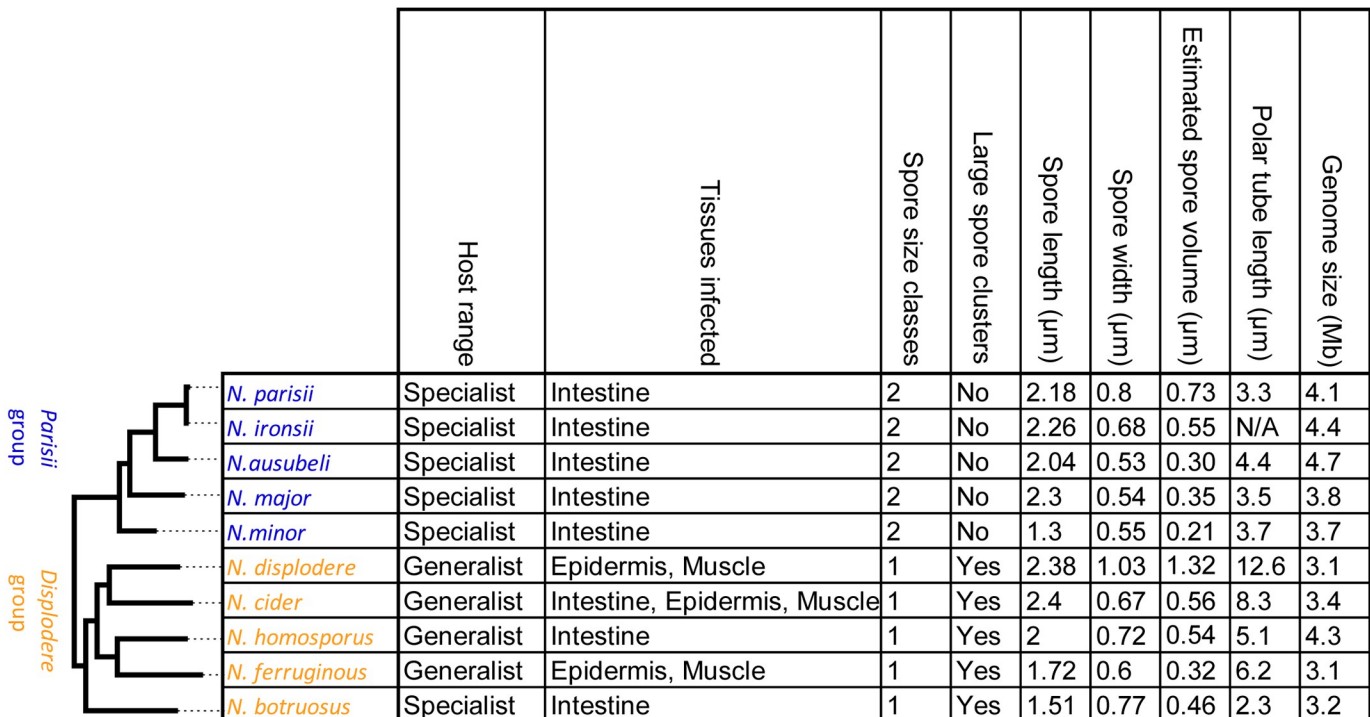

| | Host range | Tissues infected | Spore size classes | Large spore clusters | Spore length (µm) | Spore width (µm) | Estimated spore volume (µm) | Polar tube length (µm) | Genome size (Mb) |
|---|---|---|---|---|---|---|---|---|---|
| *N. parisii* | Specialist | Intestine | 2 | No | 2.18 | 0.8 | 0.73 | 3.3 | 4.1 |
| *N. ironsii* | Specialist | Intestine | 2 | No | 2.26 | 0.68 | 0.55 | N/A | 4.4 |
| *N.ausubeli* | Specialist | Intestine | 2 | No | 2.04 | 0.53 | 0.30 | 4.4 | 4.7 |
| *N. major* | Specialist | Intestine | 2 | No | 2.3 | 0.54 | 0.35 | 3.5 | 3.8 |
| *N.minor* | Specialist | Intestine | 2 | No | 1.3 | 0.55 | 0.21 | 3.7 | 3.7 |
| *N. displodere* | Generalist | Epidermis, Muscle | 1 | Yes | 2.38 | 1.03 | 1.32 | 12.6 | 3.1 |
| *N. cider* | Generalist | Intestine, Epidermis, Muscle | 1 | Yes | 2.4 | 0.67 | 0.56 | 8.3 | 3.4 |
| *N. homosporus* | Generalist | Intestine | 1 | Yes | 2 | 0.72 | 0.54 | 5.1 | 4.3 |
| *N. ferruginous* | Generalist | Epidermis, Muscle | 1 | Yes | 1.72 | 0.6 | 0.32 | 6.2 | 3.1 |
| *N. botruosus* | Specialist | Intestine | 1 | Yes | 1.51 | 0.77 | 0.46 | 2.3 | 3.2 |

*Parisii* group

*Displodere* group

**Fig 7. *Nematocida* phenotypic properties.** Table of phenotypic properties for *Nematocida* species, arranged according to the phylogenetic tree (left) determined in Fig 1. Host range, tissues infected, spore size classes, presence of large spore clusters, spore size, and polar tube length was determined in this paper and from several previously published studies [18–20]. Estimated spore volume was calculated for spore length and width measurements as previously described [17]. N/A, Not available.

contrast, all of the species in the *Displodere* group, except for *N. botruosus*, have a more generalist host range, suggesting that this property was either lost in *N. botruosus*, or gained after species divergence. Additionally, the *Displodere* group species, *N. displodere*, *N. cider*, and *N. ferruginous*, infect other tissues besides the intestine, suggesting that the ability to infect multiple tissues was possibly present after the divergence of *N. botruosus*, and then lost in *N. homosporus*.

We observed that spore size is variable throughout *Nematocida* species, with both *Nematocida* groups containing species (*N. minor* and *N. botruosus*) exhibiting some of the smallest spores. The extent of spore size variation between species exceeds the less than 10% variation that has been reported between strains of the same species [19]. Additionally, we observed that two classes of spores are present only in the *Parisii* group and large spore clusters were only observed in the *Displodere* group. Finally, we observe variability of polar tube length ranging from 2.3–12.6 µm. Our data suggests that the last common ancestor of *Nematocida* likely had a relatively short polar tube (less than ~ 4 µm). An increase in polar tube length then occurred in the *Displodere* group after the divergence of *N. botruosus*. This increase in polar tube length correlates with tissue specificity, with the three species with the longest polar tubes all infecting tissues besides the intestine. These results provide further support for microsporidian tube length being a factor in determining which tissues are infected and provide a resource to further understand molecular evolution of polar tube length once polar tube proteins in

*Nematocida* are identified [17,28]. Together, our results support evolutionary change in both morphological features and host infection in *Nematocida*, notably with loss and gain of tissue specificity.

## Discussion

To understand genomic evolution of nematode-infecting microsporidia, we sequenced the genomes of nine additional species. We then analyzed these genomes along with four previously sequenced species. We show that core cellular processes, including metabolism, are largely conserved (as determined by presence of proteins) between genera. Our analysis demonstrates that *Enteropsectra* and *Pancytospora* form a sister group to the *Enterocytozoon*, but have not undergone decay of the glycolytic pathway that has occurred in *Enterocytozoon* species [15]. This conservation of metabolism suggests that host metabolic requirements are similar between species and is in contrast to proteins predicted to interface with hosts, which are quite different between different groups of microsporidia [21]. Although we present the largest comparison of protein function between microsporidia species so far, detecting protein function from sequence alone in microsporidia is challenging. Additional analyses taking advantage of advances in structural prediction are likely to detect highly diverged proteins that would be missed based on sequence similarity alone [10,47,48].

One class of proteins that contributes to diversity between microsporidian species are the large gene families. Here we show that one of these large gene families, NemLGF1, is present in both *Nematocida* and *Pancytospora* species. There are three possibilities that could explain this observation. One, the protein family in the two groups of species could have evolved *de novo* independently of one another. This type of convergent evolution is not common, but has been observed with some enzymes that catalyze the same reaction. Comparisons of these candidate examples show that in most cases the structural fold is different and in the few cases of the fold being the same, there were structural differences with the RMSD values being greater than 3 Å [49]. As this type of protein evolution is rare and given the high predicted structural similarities that we observe, this scenario is unlikely [50]. Two, this protein family could have existed in the last common ancestor of *Nematocida* and other microsporidia, and then was lost in all lineages except for *Pancytospora* and part of *Nematocida*. From the genomes we analyzed, this would mean that this family would have been lost 7–9 times (*Nematocida Displodere* group, *N. minor*, *V. corneae*, *E. breve*, Enterocytozoon, Nosematida, Glugeida, Neopereziida, and Amblyosporida) [34]. Although this explanation is also unlikely, additional sequencing of microsporidia genomes could provide more support for or against this scenario. Three, this family was horizontally transferred between a species in each genus, though which genus was the recipient, and which was the donor, is not clear. Horizontal gene transfer is increasingly being appreciated as having a role in the evolution of eukaryotic microbes and we believe this to be the most likely explanation [51]. We also identified a family (NemLGF26) present in *N. ferruginous* which shares similarity to the previously described InterB family found in *Encephalitozoon* species [37]. Although to our knowledge, horizontal transfer of genes between microsporidia species has not been previously reported, co-infections of multiple species are often observed, suggesting that these types of events may be more common than previously appreciated [52].

Microsporidia have been shown to display much variability in their phenotypic traits [17]. However, determining how traits vary between related species has been challenging as most phylogenetic trees are built from 18S rRNA sequences, resulting in phylogenies that are often of low accuracy [3]. Here, we use genome assemblies to build a high-quality phylogeny for the *Nematocida*. By mapping several phenotypic properties onto this tree, we observe examples of

loss and gain of infection properties. As more microsporidian genomes are sequenced, this approach is likely to be useful for further understanding the evolution of microsporidia infection properties and spore characteristics. With our data we show that a generalist host range emerged in the *Displodere* group after the divergence of *N. botruosus*. One potential caveat to our measurements of host range is that they are only done with a single strain for each microsporidia and host species [16]. However, we see similar results to what was reported with other *N. parisii* and *N. ausubeli* strains, and nematode-infecting microsporidia have been observed to often infect different host strains at similar levels [19]. Previously, the size of generalist microsporidia genomes were suggested to be smaller than genomes of species that had a more narrow host range [14]. Although the generalist *Nematocida* have some of the smallest genome sizes, they are about the same size as *N. botruosus*, which has a narrow host range. This suggests that other forces can be involved in sculpting microsporidia genomes.

Although *C. elegans* has become an important model in which to study many aspects of microsporidia infection, there are currently no tools for genetically perturbing microsporidia that infect this host [53]. Knockdown of genes using RNAi has been demonstrated in microsporidia that infect honey bees, silkworms, fish, and locusts and is potentially a powerful method to investigate the function of microsporidia proteins, including those that are secreted or surface exposed [54–59]. So far, this approach has not been amenable to species that infect *C. elegans*, as *N. parisii* and the other *Nematocida* don't encode the machinery necessary to carry out RNAi [3]. Here we report that *P. epiphaga*, which infects *C. elegans* [19,26], encodes this machinery, suggesting RNAi may be possible in this species. *C. elegans* may be particularly useful for microsporidian RNAi knockdown, as RNAi of genes in *C. elegans* is highly efficient and can be carried out by feeding bacteria expressing double stranded RNA against the target transcript, resulting in dozens of successful whole genome screens [60]. Additionally, *P. epiphaga* is more closely related to several human pathogens than *N. parisii* and our genomic data will provide a useful resource for further investigations using this species.

## Material and methods

### Isolation of infected animals

Sampling of nematodes was done similar to as described previously [18]. Briefly, samples of rotting fruit or vegetation were placed on 6-cm NGM plates seeded with 10x saturated cultures of OP50-1 for 16–48 hours. Rotting matter was then removed. *N. ferruginous* was identified using differential interference contrast microscopy to identify animals displaying meronts and/or spores. *N. cider* and *N. botruosus* were identified by growing animals at 21–23˚C, washing animals off plates in M9/0.1% Tween-20, and fixing animals in 800 μl acetone. Samples were then stained with DY96 staining solution (M9, 0.1% SDS, and 20 μg/ml DY96) for 30 min. Samples were resuspended in EverBrite Mounting Medium (Biotium #23002) with DAPI and placed onto microscope slides. Microsporidia-containing animals were identified as those displaying DY96-stained spores. Nematode host species were determined using sequencing primers as described previously [61].

### Removal of contaminating bacteria from microsporidia-infected nematodes

Infected nematodes isolated in this work and previously [19] were grown on 10-cm NGM plates seeded with 10x OP50-1 until populations of worms had ingested all available bacteria. Animals were then washed off plates with M9/0.1% Tween-20 and frozen at -80˚C. 2.0 mm zirconia beads were then added and samples vortexed at 3,000 for 5 min in a bead disrupter. The

supernatant was then placed onto 6-cm NGM plates containing antibiotics (50 μg/ml carbenicillin, 25 μg/ml kanamycin, 12.5 μg/ml tetracycline, 37.5 μg/ml chloramphenicol, 200 ug/ml cefotaxime, and 100 μg/ml gentamycin) for 24–48 hours. Either the native nematode host strain that the microsporidia species was isolated in or *C. elegans* N2 animals were used to cultivate each species (See S1 Table). Mixed populations of nematodes were bleached (~4% sodium hypochlorite and 1 M NaOH) for 2–3 minutes to extract embryos which were hatched at 21˚C for 18–24 hours. L1 stage worms were then added to the antibiotic containing plates along with 10X OP50-1. After several days animals were chunked onto new NGM plates containing antibiotics. After several days animals were chunked onto NGM plates without antibiotics. If no contamination was detected, then samples were used to prepare spores. If not, the above procedure was repeated until there was no contamination.

## Preparation and DNA sequencing of microsporidian spores

Spores were prepared similar to as previously described [22]. Briefly, contaminate-free, infected populations of worms were chunked onto 12–48 10-cm OP50-1 seeded NGM plates and grown at 21˚C for 5–10 days. After worms had consumed all the OP50-1 bacteria, worms were washed off plates with M9 and frozen at -80˚C. Spores were extracted from worms as described above and then filtered through a 5 μm filter (Millipore). The concentrations of spore extracts were determined by staining spores with DY96 and counting spores on a sperm counting slide (Cell-VU) using a 20X objective on a Axio Imager.M2 (Zeiss). DNA was extracted using a MasterPure yeast DNA purification kit (Lucigen) according to manufacturer instructions using 10–50 million spores per microsporidia species. DNA libraries were prepared by The Centre for Applied Genomics (TCAG) and each species was sequenced using 2 lanes of a NovaSeq 6000 SP flow cell to produce 250 bp paired-end reads.

## Genome assembly & annotation

Adapter sequences ('AGATCGGAAGAG') were removed from paired-end raw reads using cutadapt v21 [62]. Leading and trailing low quality (quality < 3) or N bases were removed, low quality bases (quality < 30, except for *P. epiphaga* where 36 was used) were cut using a 4-base sliding window and reads less than 36 bases were dropped using Trimmomatic v0.36 [63]. Reads were mapped to the corresponding host genome (except for *N. botruosus* which was mapped to *C. elegans)* as well as the *Escherichia coli* genome using bowtie2 v2.3.4.1 [64]. Reads were assembled into contigs and scaffolds using Abyss v2.02 [65] with a Kmer of 128. Only scaffolds greater than 500 bp were retained. Assemblies were then filtered for contaminants (bacteria and host) using BlobTools v1.0.1 [66]. For some samples (S2 Table), subsampling was done prior to removing contaminants with BlobTools. Each assembly was run through Redundans v0.14a [67] with a minimum identity value of 0.85, to remove duplication. Because the laboratory the microsporidia species were cultured in regularly uses *N. parisii* ERTm1 for experiments, there exists the possibility of minor contamination of other microsporidia genomes. To correct for this, assemblies were filtered for ERTm1 [5] contamination if they met the following criteria: 99% identity and 80% query coverage. Additional filtering was done for size and coverage: scaffolds less than 750 bp were removed and those between 750 bp– 1 kbp were removed if they had less than 5X median coverage of the largest 5 scaffolds. NCBI nucleotide blast (BLASTn) [68] was used to filter remaining scaffolds that had >98% identity and >50% query cover to ERTm1/ERTm2 assemblies. Manual filtering of select scaffolds not captured in previous filtering steps was done for six of the samples (See S2 Table). Average nucleotide identity was calculated using OrthoANI 0.5.0 [69]. Protein coding genes were predicted using Prodigal v2.6.3 [70] using translation table 1. Proteins with less than 100 amino

acids were kept only if they had a significant (E = 0.001) hit to the Uniref90 protein database or Pfam hit using the hmmscan function in HMMER package v3.1b2 [71] and E-value cutoff of 0.001. The final list of proteins were annotated with Pfam [72] domains using hmmscan, signal peptides using SignalP v4.1f [73], and transmembrane domains using TMHMM v2.0c [74]. Functional annotation of proteins was also done using BlastKOALA [75] (S3 Table).

## Phylogeny and genome characterization of microsporidia species

The phylogenetic tree (Fig 1A) was generated using 45 species. Species whose proteome was downloaded from NCBI are listed in S5 Table. The remaining proteomes were predicted from the genomes assembled in this study. In addition, proteins were predicted, as described above, from two assemblies available on NCBI for the following species: *Metchnikovella incurvata*, and *Enteropsora canceri*. Orthofinder v2.5.2 [32] was used with -M msa option to obtain a MSA species tree with a minimum of 77.8% of species having single copy genes in an orthogroup. Tree was rooted using *Rozella Allomycis* [76] in FigTree v1.4.4. Protein content, genome size, as well as genome completeness statistics were computed for each species using BUSCO v3.1.0 [77] and microsporidia odb9.

## Pathway conservation in microsporidia species

InterProScan v5.51–85.0 [78] was run on all species with -goterms option. A database was made with Pombe GO-slim terms (https://www.ebi.ac.uk/QuickGO/slimming) and their descendants (found using QuickGO API using the 'find descendants' option, https://www.ebi.ac.uk/QuickGO/api/index.htmllar#!/gene_ontology/). For each species, protein to GO-term annotations from InterProScan were mapped to the above-mentioned database while retaining count of proteins mapping to each GO term. Since multiple GO terms map to a single GO slim, a unique protein count was obtained for each GO slim. Using *Rozella allomycis* as the base species, a subset of the Pombe GO-slim terms were obtained and the subset was then searched against the remaining species to compare pathway conservation across species.

## Identifying large gene families

OrthoMCL v2.0.9 [79] was run using all proteomes and with the following configuration settings: percentMatchCutoff = 50 and evalueExponentCutoff = -5, suggested minimum protein length and maximum percent stop codons, and MCL inflation value of 1.5. Paralogous orthogroups were extracted (at least 10 members in a single genome). The genome with the highest number of paralogs for a specific gene family was used as the seed sequences to expand gene families. Seed sequences were searched against each microsporidia genome iteratively using the jackhmmer function in HMMER and an E-value cutoff of 10–5. All resulting large gene families were assessed for signal peptides, transmembrane domains, and Pfam domains. A family was classified as having signal peptides if >50% of the proteins in the family had a signal peptide. The same criteria were applied for transmembrane domains. Families that did not contain either signal peptides or transmembrane domains in greater than >50% of proteins were not included in our analysis. All families with 90% or greater protein overlap were collapsed, with the exception of NemLGF2 and NemLGF20 which were collapsed with less than 90% overlap in some cases. We used OrthoMCL to assign membership with families with >5% Pfam representation in either LRR, ZF, or peptidase domains [21]. To determine if these large gene families were present in other microsporidia species, we searched the microsporidia genomes in Fig 1 with family-specific HMM models using HMMER. Protein sequences that had at least an E-value of 10–5 were considered as belonging to that family.

## Structural modeling of large gene families

Models of members of large gene families were generated using AlphaFold as implemented in ColabFold [38,39] (https://colab.research.google.com/github/sokrypton/ColabFold/blob/main/AlphaFold2.ipynb). Default settings were used for NEPG_02057, and for the other proteins PSI-BLAST [68] was used to identify members and multiple sequence alignments were generated using HHblits (https://toolkit.tuebingen.mpg.de/tools/hhblits) [80]. All the models analyzed have overall pLDDT scores over 85. Alignments of NemLGF1 domains were performed using PyMOL v2.5.3 (https://pymol.org). Structural similarity to solved structures was determined using Dali (http://ekhidna2.biocenter.helsinki.fi/dali/) [81] Statistical significance for Dali was defined as a Z-score greater than 10 with over half of the query residues aligned. Structural similarity to predicted structures from AlphaFold was determined using Foldseek (https://search.foldseek.com/search) [41]. Statistical significance for Foldseek was defined as an E-value less than 0.05. The percent identity between proteins was determined using MUSCLE (https://www.ebi.ac.uk/Tools/msa/muscle/) [82].

## Host specificity assays

The following nematode species (strains) were used for infection assays: *C. elegans* (N2), *Caenorhabditis briggsae* (JU2507), *Caenorhabditis nigoni* (JU1422), *Caenorhabditis remanei* (JU2796*)*, *Caenorhabditis tropicalis* (JU1373), *Oscheius tipulae* (JU1505), and *Panagrellus* sp. 2 (AWR79). Mixed populations of nematodes were maintained on 10-cm seeded NGM plates for at least three generations without starvation at 21˚C. Animals were then washed off plates and embryos extracted with bleach as described above, except for *Panagrellus* sp. 2. Synchronized animals of *Panagrellus* sp. 2 were prepared by washing animals with M9 three times and then filtering through a 20 μm nylon filter (Millipore SCNY00020). This filtering was then repeated a second time. 400 L1 stage animals of each species were infected with the following microsporidia species (strain [spore amount]): *N. parisii* (JUm1248[3 million]), *N. ausubeli* (ERTm6[0.8 million]), *N. cider* (AWRm77[4 million]), *N. ferruginous* (LUAm3[20 or 40 million]), and *N. botruosus* (AWRm80[28 million]). These doses were chosen so that average percentage infection was between 80–99% in either *C. elegans* or the host species the microsporidia was found in, with the exception of *N. ferruginous* where this level of infection was not possible to achieve. Animals and spores were mixed with 400 μl 10X OP50-1 and placed onto 6-cm NGM plates. Plates were dried in a clean cabinet and incubated at 21˚C for 96 hours. Animals were then washed off plates, fixed, stained, and placed onto slides as described above. The percentage of animals infected was determined by counting the number of P0 animals that contained newly formed clusters of microsporidia spores.

## Microsporidia spore size measurements

Infected, fixed, and stained animals were prepared as described above. Images of these infected animals were taken using an 63X objective on an apotome-equipped Axio Imager.M2 (Zeiss). Images were taken as z-stack maximum intensity projections. These images were analyzed using the straight line tool in ImageJ 1.52 [83] to measure the length and width of each spore. Only spores that were not immediately adjacent to other spores were measured. At least 40 spores were measured for each species from a minimum of two infected worms. Presence of large spore clusters was based on visual inspection of either DY96-stained spores in S15 Fig or from previously published images [19].

## Infection of *aaim-1* mutants

N2 and *aaim-1* mutants (kea22 and kea28) [23] were infected with *N. cider* (4 million spores) as described above. To quantify pathogen burden within animals (DY96), regions of interest were used to outline individual worms followed by subjection to the "threshold" followed by "measure" tools in FIJI [84].

## Tissue specificity determination

Strains of *C. elegans* each expressing GFP specifically in the intestine (ERT413), epidermis (ERT446), or muscle (HC46) were grown on standard NGM plates seeded with OP50-1 to the gravid adult stage. Gravid adults were then harvested in M9 and transferred to 15 mL conical tubes, washed three times to reduce bacterial excess, and pelleted by centrifuging at 3000 x g. The pellets were resuspended in 1 mL of M9 buffer and added with 1 mL of bleach solution (1:4 volume ratio of 5M NaOH: 6% NaClO; final concentration of 0.5M NaOH, 2.4% NaClO), incubated for 2 min at room temperature, and washed three times with 13 mL of M9. After the final wash, the solutions containing embryos were incubated overnight on a rotator at room temperature to synchronize to the L1 stage.

~1000 L1 stage animals of each strain were plated onto a 6-cm NGM plate with either 1 million *N. ferruginous* or *N. cider* spores. Infection plates were allowed to dry and kept at 20°C. At 3 days post infection, animals were harvested in M9 buffer and fixed with 4% paraformaldehyde in PBST (1x PBS, 0.1% Tween-20) at room temperature for 40 min on a rotator for fluorescent in situ hybridization. Fixed animals were then washed four times, each with 1 mL of PBST, then incubated overnight with a mixture of CAL Fluor Red 610 (CF610)-tagged microsporidian probes, including MicroA-CF610 (CTCTGTCCATCCTCGGCAA), MicroB-CF610 (CTCTCGGCACTCCTTCCTG), MicroD-CF610 (CGAAGGTTTCCTCGGATGTC), and MicroF-CF610 (AGACAAATCAGTCCACGAATT), each at a final concentration of 2.5 ng/mL in hybridization buffer (900 mM NaCl, 20 mM Tris pH 7.5, 0.01% SDS) on a thermal shaker at 46°C, 1000 rpm. After hybridization, animals were washed with FISH wash buffer (hybridization buffer, 5mM EDTA) for three times, each for 30 min, on a thermal shaker at 48°C, 1000 rpm. After washing, animals were rinsed once with PBST, mounted, and imaged with confocal microscope Leica Stellaris 5. Each sample was scanned randomly for 30 infected worms and images were taken in different planes to examine all the infection areas in a single worm. For each strain, infected animals were binned into "yes" or "no" respective to the fluorescent tissue. An infected animal was considered to have a specific tissue type infected if there was at least one infection area colocalized with the tissue marker.

## *N. cider* and *N. ferruginous* assays to measure change in host color

~1000 N2 animals at L1 stage on a 10-cm NGM plate were infected with 3 million spores of either *N. ferruginous* or *N. cider*. As the animals grew, plates were monitored to add more food (1 mL of 10x OP50-1 concentrated from a culture of $OD_{600}$ of 0.5–0.7) to prolong the infection in the population. After 7–10 days post infection, some animals appeared brown compared to other animals on the same plate and uninfected control under standard dissection microscope. Control animals and brown, infected animals were picked and mounted onto a 2% agarose pad and imaged with Eclipse Ni microscope (Nikon). All images were taken at the same light intensity and white balanced.

## Measurements of polar tube lengths

Polar tube lengths were measured for *N. parisii* (ERTm1), *N. ausubeli* (ERTm2), N. minor (JUm1510), N. major (JUm2507), *N. homosporus (*JUm1504*)*, *N. cider* (AWRm77), *N. ferruginous* (LUAm3), and *N. botruosus* (AWRm80). To encourage polar tube extrusion, microsporidia spores in microcentrifuge tubes were exposed to two freeze thaw cycles, after the initial thaw from -80°C. This was done by freezing at -80°C for 10 minutes followed by thawing at room temperature for 10 minutes. Nile red (Sigma-Aldrich 72485) (1mg/ml in acetone) was prepared, sequentially diluted to 100 µg/ml in M9, and added to a final concentration of 10 µg/ml in freeze-thaw treated spores. Spores were incubated with Nile red for 30 minutes in the dark followed by 2 µl of Calcofluor White (Sigma-Aldrich 18909). Polar tubes were imaged at 63x magnification using an Axio Imager.M2 (Zeiss). Polar tube measurements were performed on FIJI [84], using the freehand line tool to trace polar tubes followed by selecting the Analyze ➔ Measure option.

## Taxonomic summaries

Phylum: Microsporidia Balbiani 1882

Species: *Nematocida cider* n. sp. Wadi et al. 2022.

The type strain AWRm77 was found in 2017 inside of its type host nematode *Caenorhabditis* sp. 8 strain AWR77, which was isolated from rotting apples in Stow, Massachusetts, United States. The type material is deposited in the collection of the corresponding author (AWR). The genome of this microsporidia strain has been sequenced and deposited in Genbank under accession JALPNA000000000. This is a novel species based on the divergence of the genome from other sequenced species. This species has been observed to infect the intestinal, muscle and epidermal tissues of *C. elegans*. Experimental infection has been observed in *C. elegans* (N2), *Caenorhabditis briggsae* (JU2507), *Caenorhabditis nigoni* (JU1422), *Caenorhabditis remanei* (JU2796*)*, *Caenorhabditis tropicalis* (JU1373), *Oscheius tipulae* (JU1505), and *Panagrellus* sp. 2 (AWR79). Transmission occurs horizontally. The spores are ovoid and measure as 2.4 x 0.67 µm. The fired polar tube measures 8.3 µm. This species is named after the brown color the animals display after being infected for several days and this color resembles that of cider donuts available in the area where the specimen was found.

## Species: *Nematocida ferruginous* n. sp. Wadi et al. 2022

The type strain LUAm1 was found in 2017 inside of its type host nematode *C. elegans* strain LUA1, which was isolated from a rotting *Heracleum* sp. stem in Santeuil, France (GPS coordinates 49° 7' 16.99" N, 1° 57' 3.918" E). The type material is deposited in the collection of the corresponding author (AWR). The genome of this strain has been sequenced and deposited in Genbank under accession JALPMW000000000. This is a novel species based on the divergence of the genome from other sequenced species. This species has been observed to infect the muscle and epidermal tissues of *C. elegans*. Experimental infection has been observed in *C. elegans* (N2), *Caenorhabditis briggsae* (JU2507), *Caenorhabditis remanei* (JU2796*)*, *Caenorhabditis tropicalis* (JU1373), *Oscheius tipulae* (JU1505). Transmission occurs horizontally and is likely via a fecal-oral route. The spores are ovoid and measure as 1.72 x 0.6 µm. The fired polar tube measures 6.2 µm. This species is named after the rust-like brown color the animals display after being infected for several days.

## Species: *Nematocida botruosus* n. sp. Wadi et al. 2022

The type strain AWRm80 was found in 2018 inside of its type host *Panagrellus* sp. 2 strain AWR80, which was isolated from rotting apples in Georgina, Ontario, Canada. The type

material is deposited in the collection of the corresponding author (AWR). The genome of this strain has been sequenced and deposited in Genbank under accession JALPMX000000000. This is a novel species based on the divergence of the genome from other sequenced species. This species has only been observed to infect the intestine of this animal and transmission occurs horizontally. The spores are ovoid and measure as 1.51 x 0.76 μm. The fired polar tube measures 2.3 μm. This species was named after the prominent large spore clusters formed.

## Supporting information

**S1 Fig. Pairwise nucleotide identity of nematode-infecting microsporidia genomes.** The pairwise nucleotide identity of the three *N. ferruginous* assemblies and the other nematode-infecting microsporidia genomes were calculated and displayed as a heat map.
(TIF)

**S2 Fig. Phylogenetic tree of *Nematocida* species.** The phylogeny of 10 *Nematocida* species was determined from single-copy orthologs identified using OrthoMCL. Phylogenetic tree was generated using RaxML. *M. daphniae* is shown as an outgroup. Bootstrap values are indicated at each node. Scale indicates changes per site.
(PDF)

**S3 Fig. Genome size and number of encoded proteins are correlated in nematode-infecting microsporidia.** The correlation between genome size and protein number in 13 nematode-infecting microsporidia genomes is shown as a scatter plot. Pearson correlation coefficient and p-value are shown in the top left.
(PDF)

**S4 Fig. Conservation of carbohydrate metabolic processes across microsporidia genomes.** Membership of proteins from *R. allomycis* and 40 microsporidia species in descendant GO terms from the Pombe GO-slim category "carbohydrate metabolic process" was determined. The number of proteins from each species determined to belong to each Go term is shown as a heatmap with GO-slim categories in rows and microsporidia species in columns. Only descendant GO terms that contain at least one protein from any of these species is shown. Legend for the number of proteins in each cell is shown at the right. Phylogenetic tree, shown at bottom, was constructed using Orthofinder. Several species (*Pseudoloma neurophilia*, *Dictyocoela roeselum*, *Cucumispora dikerogammari*, and *Nosema apis*) were excluded due to poorer quality genome assemblies (See Fig 1). *Nematocida* species are highlighted with a blue box. *Enteropsectra* and *Pancytospora* species are highlighted with a green box.
(PDF)

**S5 Fig. Conservation of generation of precursor metabolites and energy pathways across microsporidia genomes.** Membership of proteins from *R. allomycis* and 40 microsporidia species in descendant GO terms from the Pombe GO-slim category "generation of precursor metabolites and energy" was determined. The number of proteins from each species determined to belong to each Go term is shown as a heatmap with GO-slim categories in rows and microsporidia species in columns. Only descendant GO terms that contain at least one protein from any of these species is shown. Legend for the number of proteins in each cell is shown at the right. Phylogenetic tree, shown at bottom, was constructed using Orthofinder. Several species (*Pseudoloma neurophilia*, *Dictyocoela roeselum*, *Cucumispora dikerogammari*, and *Nosema apis*) were excluded due to poorer quality genome assemblies (See Fig 1). *Nematocida* species are highlighted with a blue box. *Enteropsectra* and *Pancytospora* species are highlighted with a green box.
(PDF)

**S6 Fig. Conservation of tRNA metabolic processes across microsporidia genomes.** Membership of proteins from *R. allomycis* and 40 microsporidia species in descendant GO terms from the Pombe GO-slim category "tRNA metabolic process" was determined. The number of proteins from each species determined to belong to each Go term is shown as a heatmap with GO-slim categories in rows and microsporidia species in columns. Only descendant GO terms that contain at least one protein from any of these species is shown. Legend for the number of proteins in each cell is shown at the right. Phylogenetic tree, shown at bottom, was constructed using Orthofinder. Several species (*Pseudoloma neurophilia*, *Dictyocoela roeselum*, *Cucumispora dikerogammari*, and *Nosema apis*) were excluded due to poorer quality genome assemblies (See Fig 1). *Nematocida* species are highlighted with a blue box. *Enteropsectra* and *Pancytospora* species are highlighted with a green box.
(PDF)

**S7 Fig. Conservation of cellular amino acid metabolic processes across microsporidia genomes.** Membership of proteins from *R. allomycis* and 40 microsporidia species in descendant GO terms from the Pombe GO-slim category "cellular amino acid metabolic process" was determined. The number of proteins from each species determined to belong to each Go term is shown as a heatmap with GO-slim categories in rows and microsporidia species in columns. Only descendant GO terms that contain at least one protein from any of these species is shown. Legend for the number of proteins in each cell is shown at the right. Phylogenetic tree, shown at bottom, was constructed using Orthofinder. Several species (*Pseudoloma neurophilia*, *Dictyocoela roeselum*, *Cucumispora dikerogammari*, and *Nosema apis*) were excluded due to poorer quality genome assemblies (See Fig 1). *Nematocida* species are highlighted with a blue box. *Enteropsectra* and *Pancytospora* species are highlighted with a green box.
(PDF)

**S8 Fig. Conservation of lipid metabolic processes across microsporidia genomes.** Membership of proteins from *R. allomycis* and 40 microsporidia species in descendant GO terms from the Pombe GO-slim category "lipid metabolic process" was determined. The number of proteins from each species determined to belong to each Go term is shown as a heatmap with GO-slim categories in rows and microsporidia species in columns. Only descendant GO terms that contain at least one protein from any of these species is shown. Legend for the number of proteins in each cell is shown at the right. Phylogenetic tree, shown at bottom, was constructed using Orthofinder. Several species (*Pseudoloma neurophilia*, *Dictyocoela roeselum*, *Cucumispora dikerogammari*, and *Nosema apis*) were excluded due to poorer quality genome assemblies (See Fig 1). *Nematocida* species are highlighted with a blue box. *Enteropsectra* and *Pancytospora* species are highlighted with a green box.
(PDF)

**S9 Fig. Conservation of sulfur compound metabolic processes across microsporidia genomes.** Membership of proteins from *R. allomycis* and 40 microsporidia species in descendant GO terms from the Pombe GO-slim category "sulfur compound metabolic process" was determined. The number of proteins from each species determined to belong to each Go term is shown as a heatmap with GO-slim categories in rows and microsporidia species in columns. Only descendant GO terms that contain at least one protein from any of these species is shown. Legend for the number of proteins in each cell is shown at the right. Phylogenetic tree, shown at bottom, was constructed using Orthofinder. Several species (*Pseudoloma neurophilia*, *Dictyocoela roeselum*, *Cucumispora dikerogammari*, and *Nosema apis*) were excluded due to poorer quality genome assemblies (See Fig 1). *Nematocida* species are highlighted with a blue

box. *Enteropsectra* and *Pancytospora* species are highlighted with a green box.
(PDF)

**S10 Fig. Conservation of mRNA metabolic processes across microsporidia genomes.** Membership of proteins from *R. allomycis* and 40 microsporidia species in descendant GO terms from the Pombe GO-slim category "mRNA metabolic process" was determined. The number of proteins from each species determined to belong to each Go term is shown as a heatmap with GO-slim categories in rows and microsporidia species in columns. Only descendant GO terms that contain at least one protein from any of these species is shown. Legend for the number of proteins in each cell is shown at the right. Phylogenetic tree, shown at bottom, was constructed using Orthofinder. Several species (*Pseudoloma neurophilia*, *Dictyocoela roeselum*, *Cucumispora dikerogammari*, and *Nosema apis*) were excluded due to poorer quality genome assemblies (See Fig 1). *Nematocida* species are highlighted with a blue box. *Enteropsectra* and *Pancytospora* species are highlighted with a green box.
(PDF)

**S11 Fig. Conservation of nucleobase-containing small molecule metabolic processes across microsporidia genomes.** Membership of proteins from *R. allomycis* and 40 microsporidia species in descendant GO terms from the Pombe GO-slim category "nucleobase-containing small molecule metabolic process" was determined. The number of proteins from each species determined to belong to each Go term is shown as a heatmap with GO-slim categories in rows and microsporidia species in columns. Only descendant GO terms that contain at least one protein from any of these species is shown. Legend for the number of proteins in each cell is shown at the right. Phylogenetic tree, shown at bottom, was constructed using Orthofinder. Several species (*Pseudoloma neurophilia*, *Dictyocoela roeselum*, *Cucumispora dikerogammari*, and *Nosema apis*) were excluded due to poorer quality genome assemblies (See Fig 1). *Nematocida* species are highlighted with a blue box. *Enteropsectra* and *Pancytospora* species are highlighted with a green box.
(PDF)

**S12 Fig. Signal peptides and transmembrane domains in nematode-infecting microsporidia genomes.** The percentage of proteins in each genome predicted to contain either signal peptides or transmembrane domains is shown.
(TIF)

**S13 Fig. Structural models of nematode-infecting microsporidia large gene families.** (A-B) AlphaFold models of *N.* parisii NEPG_02057 (B) and *N. ausubeli* NERG_01890 (B). (C-E) Aligned structures of the N-terminal (C), middle (D), and C-terminal (E) domains. N, N-terminal. M, middle. C, C-terminal. (F-I) AlphaFold models of PanLGF1 member PAPHI01_0471 (F), NemLGF11 NEMIN01_1006 (G), EbrLGF1 ENBRE01_2406 (H), and NemLGF2 NEDG_02234 (I).
(JPG)

**S14 Fig. *N. ferruginous* infects multiple host species.** Six species of L1 stage nematodes were infected with either 20 (replicate 1) or 40 million (replicate 2) *N. ferruginous (*LUAm3) spores. After 96 hours of incubation with spores, animals were fixed and stained with DY96. Percent of each population of animals infected with each species of microsporidia. Data is displayed as a heat map with host species in rows, each *N. ferruginous* replicate in columns, and the value of each cell being the percent of each population that displayed newly formed microsporidia spores. Legend is displayed at the right. 50–196 animals were counted for each sample.
(TIF)

**S15 Fig. Spore formation observed in the intestine for *N. botruosus* and throughout the animal for *N. cider* and *N. ferruginous*. (A-C)** L1 stage animals were exposed to spores and incubated for 96 hours, fixed, and stained with DY96. Representative images were taken with the apotome module of a ZEISS Axio Imager at 63x magnification. Multiple z-planes were imaged, and a maximum intensity projection is displayed for each sample. Scale bars, 20 μm. **(A)** *Panagrellus* sp. 2 infected with 28 million *N. botruosus* spores. **(B)** *C. elegans* N2 infected with 4 million *N. cider* spores. **(C)** *C. elegans* N2 infected with 40 million *N. ferruginous* (LUAm3) spores. **(D)** *C. elegans* N2 animals were either infected with 3 million *N. cider* spores or 3 million *N. ferruginous* spores for 5 days. Images taken with Nikon Eclipse Ni at 10 x magnification.
(PDF)

**S16 Fig. AAIM-1 is necessary for efficient infection by *N. cider*.** N2 and *aaim-1* mutant animals infected with 4 million *N. cider* spores, fixed at 96 hours, and stained with DY96. 20–30 worms quantified per replicate. The percentage of the animal containing DY96 signal is shown. Mean ± SEM represented by horizontal bars. P-values determined via One-way Anova with post hoc. Significance defined as **** $p < 0.0001$.
(PDF)

**S1 Table. Summary of microsporidia species and their hosts.**
(XLSX)

**S2 Table. Methods used to assemble each microsporidia genome.**
(XLSX)

**S3 Table. Annotation of microsporidia genomes.**
(XLSX)

**S4 Table. Summary of genome assembly statistics.**
(XLSX)

**S5 Table. List of genome assemblies used.**
(XLSX)

**S6 Table. RNAi pathway proteins in *Pancytospora* and *Enteropsectra*.**
(XLSX)

**S7 Table. List of large gene family proteins.**
(XLSX)

**S8 Table. NemLGF26 proteins present in other microsporidia species.**
(XLSX)

**S1 Data. All infection experiment data.**
(XLSX)

## Acknowledgments

We thank Yin Chen Wan, Edward James, and Winnie Zhao for providing helpful comments on the manuscript. We thank Marie-Anne Félix for providing previously published microsporidia species and for providing support to R. L. during isolation of *N. ferruginous*. Some strains were provided by the CGC, which is funded by NIH Office of Research Infrastructure Programs (P40 OD010440).

## Author Contributions

**Conceptualization:** Lina Wadi, Hala Tamim El Jarkass, Tuan D. Tran, Robert J. Luallen, Aaron W. Reinke.

**Data curation:** Aaron W. Reinke.

**Formal analysis:** Lina Wadi.

**Funding acquisition:** Robert J. Luallen, Aaron W. Reinke.

**Investigation:** Lina Wadi, Hala Tamim El Jarkass, Tuan D. Tran, Nizar Islah, Aaron W. Reinke.

**Methodology:** Lina Wadi, Aaron W. Reinke.

**Project administration:** Aaron W. Reinke.

**Resources:** Robert J. Luallen, Aaron W. Reinke.

**Supervision:** Robert J. Luallen, Aaron W. Reinke.

**Visualization:** Lina Wadi, Hala Tamim El Jarkass, Aaron W. Reinke.

**Writing – original draft:** Lina Wadi, Aaron W. Reinke.

**Writing – review & editing:** Lina Wadi, Hala Tamim El Jarkass, Tuan D. Tran, Nizar Islah, Robert J. Luallen, Aaron W. Reinke.

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
