## [Decision Letter · Decision Letter 0]

13 Jan 2023

Dear %TITLE% Reinke,

Thank you very much for submitting your manuscript "Genomic and phenotypic evolution of nematode-infecting microsporidia" for consideration at PLOS Pathogens. As with all papers reviewed by the journal, your manuscript was reviewed by members of the editorial board and by several independent reviewers. In light of the reviews (below this email), we would like to invite the resubmission of a significantly-revised version that takes into account the reviewers' comments.

This study combines genomic, computational and phentoypic data for 9 species of microsporidia that infect nematodes. Excellent computational analyses are presented and all reviewers agree that this body of work provides a novel resource to facilitate future research on this group of parasitic microsporidia. All reviewers applaud the authors for their context investigating various phenotypic traits that differ among the microsporidia but feel that some of the experimental validation to support the conclusions drawn are underdeveloped, and this certainly needs to be addressed. Reviewer 2 lays out a compelling template how to re-work the manuscript, into a more coherent flow that integrates the various phenotypes to provide a more compelling, evidence-based dataset that supports some of the conclusions made. Alternative explanations should be raised, in particular the support for the Horizontal Gene Transfer of a specific gene family is interesting, but needs to be vetted more appropriately - it may very well be correct, but alternative explanations also exist that are not discussed. The manuscript has the potential to lay the foundation for a new field of study that will be valuable for the community, but the data needs to be more coherently integrated, with supporting evidence, which is the primary challenge to rectify prior to publication.

We cannot make any decision about publication until we have seen the revised manuscript and your response to the reviewers' comments. Your revised manuscript is also likely to be sent to reviewers for further evaluation.

Sincerely,

Michael E. Grigg

Guest Editor

PLOS Pathogens

P'ng Loke

Section Editor

PLOS Pathogens

Kasturi Haldar

Editor-in-Chief

PLOS Pathogens

orcid.org/0000-0001-5065-158X

Michael Malim

Editor-in-Chief

PLOS Pathogens

orcid.org/0000-0002-7699-2064

This study combines genomic, computational and phentoypic data for 9 species of microsporidia that infect nematodes. Excellent computational analyses are presented and all reviewers agree that this body of work provides a novel resource to facilitate future research on this group of parasitic microsporidia. All reviewers applaud the authors for their context investigating various phenotypic traits that differ among the microsporidia but feel that some of the experimental validation to support the conclusions drawn are underdeveloped, and this certainly needs to be addressed. Reviewer 2 lays out a compelling template how to re-work the manuscript, into a more coherent flow that integrates the various phenotypes to provide a more compelling, evidence-based dataset that supports some of the conclusions made. Alternative explanations should be raised, in particular the support for the Horizontal Gene Transfer of a specific gene family is interesting, but needs to be vetted more appropriately - it may very well be correct, but alternative explanations also exist that are not discussed. The manuscript has the potential to lay the foundation for a new field of study that will be valuable for the community, but the data needs to be more coherently integrated, with supporting evidence, which is the primary challenge to rectify prior to publication.

Reviewer's Responses to Questions

**Part I - Summary**

Reviewer #1: The paper by Lina Wadi describes the acquisition and analysis of multiple closely nematode-infecting microsporidia strains.

The paper is well written, and the presented analyses seems thoroughly performed.

Reviewer #2: This study reports a combination of genomic and phenotypic data on multiple genera of nematode-infecting microsporidia. It provides a novel resource to enhance future research on this recently discovered group of parasites. The characterization of Nematocida phenotypes in a phylogenetic context makes a particularly valuable contribution – microsporidia are understudied and can be challenging to work with; this study takes advantage of the experimental tractability of nematode-infecting microsporidia to measure multiple traits across the genus.

I reviewed this manuscript with a member of my lab group. We found the methods to be sound and the writing clear. We overall feel positively about the work. Below, we highlight cases where claims require more support and justification from the data and analyses. Several elements of the paper would be strengthened with more biological grounding - the discussion of large gene families for example. Finally, many of the figures are so data-rich that it’s challenging to extract the main point.

Reviewer #3: This manuscript reports and analyze genomes of nine species of nematode-infecting microsporidia, including two species in a distinct clade more closely related to human pathogens than the previously studied Nematocida, and three newly described Nematocida species. This adds considerably to the previous four Nematocida species genomes that were available, although the present assemblies are less complete. A particularly intringuing result concerns the apparent horizontal transfer in a gene family between distant genera of nematode-infecting microsporidia. The authors also provide data regarding infection properties and morphology of the three new species.

Overall, this is a useful and extensive characterization of nematode-infecting microsporidia.

**Part II – Major Issues: Key Experiments Required for Acceptance**

Reviewer #1: I found however, the paper to be very descriptive. However, I believe this paper provides some important insight into microsporidian genome evolution. I provide some feedback the authors could use to further improve impact to the entire field, as opposed to center their findings primarily on nematode microsporidians. In particular, it would be interesting to see if the patters the authors observed extrapolate to other microsporidians. In particular, there are multiple closely related species in the Encephalitozoon/Ordospora clade that have sequences available and could be used to investigate similar patterns at larger scale.

Below are some comments/suggestions.

- L126: Using OrthoFinder29, we generated a phylogenetic tree of our nine newly assembled.

Shouldn’t this be: using OF we identified Orthologous sequences, which we used to produce phylogenetic trees?

- L215 : exception of NemLGF2, which shares similarity to Leucine Rich Repeat proteins.

The presence and molecular analyses of Leucine Rich Repeats have been published before and should be not considered new. See Campbell et al. 2013 Plos Genetics for a thorough analysis of these proteins.

- L296: We observed that spore size is flexible throughout Nematocida species…

What about intra-strain? If the authors want to emphasize size difference among species, one can expect those sizes not to vary within a species. Can the authors show this?

- L3-4. These results provide further support for microsporidian tube length being a factor in determining which tissues are infected.

How about the PTP genes? Do they behave in different ways among species? For example, are these present in different copy numbers etc that could reflect the observed phenotypic variability?

- L329: …present in any other microsporidia species, the most likely explanation is that this family was horizontally transferred between a species in each genus…

Or perhaps these appeared de novo?

HGT is testable and if it happened here, one would expect a very high sequence identity in these proteins among the clades. Is this being observed here? Any evidence these genes evolve differently than others?

Reviewer #2: 1. Connection between claims and data/analyses: There are results presented in the Abstract as major conclusions that we felt were not sufficiently supported in the main text.

A) The suggestion of horizontal gene transfer between distantly related nematode-infecting microsporidia is a pretty big claim, especially given that there are no prior reports of horizontal transfer between microsporidia. We didn’t feel that this was an unreasonable hypothesis given the pattern observed, but we felt the data were insufficient to make horizontal gene transfer the “mostly likely explanation” (line 329) to put forward. We would like see either 1) more justification of this claim (are there other data that support this hypothesis? Including a supplemental phylogeny for other large, distributed gene families so we can make a comparison?) and/or 2) more qualification of this claim, including presentation of alternate hypotheses (undersampling of the microsporidia phylogeny?) and the type of data necessary to test them.

B) Lines 19-21 argue that these large gene families are likely used to interact with host cells. This idea is raised again in lines 79-80 of the Introduction, though with limited clarification of the prior result in the literature, then again in the Conclusion lines 325-327 stating that “these families were enriched for secretion signals and transmembrane domains.” We didn’t see this analysis of enrichment in the Results, beyond Figure 4 showing the distribution of signal peptides and transmembrane domains in expanded gene families. What is the distribution of signal peptides and transmembrane domains in a random sampling of gene families that are not expanded? Are they less likely to have these domains than expanded families?

2. The large gene families section was also an area where we felt that substantially more discussion and biological grounding was needed for the reader to understand if, and why, they matter. What are the hypotheses for why these gene families might be expanded, particularly when the major trend is gene loss? To what extent are the observed patterns consistent with those hypotheses? How sensitive are these gene family results to the specific parameters used in OrthoMCL? Can the authors justify their choice of parameters as the most appropriate for detecting large gene families in microsporidia?

We felt similarly about the protein structures – these protein structures can be modeled and presented, but what do we learn from them? We thought these models needed more justification if they were to be included, or alternately could be excluded with no impact on the paper.

3. The figures: As noted above, the figures opt to show as much information as possible, which is valuable as a resource, but makes them very hard to look at and interpret. Figure 2: can the authors somehow delineate (maybe using boxes) data associated with the nematode-infecting microsporidia? This subset of the data is currently hard to track. Figure 3: Same issue. Figure 4: this figure would be helped with a more supportive legend that unpacks the significance of the different components of the figure. Alternately, a more valuable figure would be a comparison of signal peptide and transmembrane domains in expanded vs. non-expanded gene families. Figure 5: this figure is too dense and complex. We thought B-F did not substantially contribute to the paper. For A, is there a way of simplifying these data, and modifying the color scheme, to make the pattern you’re most interested in (the potential for HGT) clearer? Figure S1: a table would be better for presentation of these data. The heat map is inappropriate given the scale of the data. Fig. S4-S11: same suggestion as for Figure 2. Fig. S12: this is too small and uses too many colors to be interpretable. We don’t feel that it’s necessary. If it does make a point that the authors think is critical, can it be simplified to highlight that point? Fig. S13: we didn’t feel that this added to the paper and the second sentence of the title is an overstatement. Fig. S14: could this be revised to show the actual data, with error estimates (either binomial if replicates are kept separate or standard error if averaged)? That would be more appropriate and might also help understand if infection rates really declined with increased dose (they appear to, but we don’t know the extent to which that arises in part from a wide confidence interval around the estimate). All that being said, we thought Figure 1 was a very nice presentation of data!

Reviewer #3: 1) This manuscript describes two new species of microsporidia. Please conform to some minimal standard of species descriptions. Deposit type specimen (live cultures of infected nematodes are fine). Make the basis for raising a new species explicit, for example genome divergence. Latinize the names. Maybe provide a light microscopy picture of spores (either isolated or in the host).

2) Phylogeny and character mapping: the authors use a parsimony logic to infer character evolution. I would be more prudent in the conclusions as there are few species in Fig. 7. For example, the claim in the abstract that the branch leading to N. homosporus lost the ability to invade multiple tissues is weak and needs to be removed.

**Part III – Minor Issues: Editorial and Data Presentation Modifications**

Reviewer #1: see some comments above.

Reviewer #2: 64 revise as “suggesting that these traits are quite evolutionarily labile in microsporidia.” Plastic has a distinct meaning that isn’t what you want here. It would also be helpful to have a little more context with regards to what is meant by “morphological and infection properties”

79-80 provide more context for this prior result, given that it is a major feature of this paper

84 to be involved

87 analysis

98 the results shed limited light on evolutionary mechanisms; a better fit would be: “Our study reveals the evolutionary relationships of nematode-infecting microsporidia and…”

104 have revealed

188-190 This sentence collides with the prior sentence (181-182) and subsequent sentence (192-194) indicating that there are several gene families that have more than 10 members present in multiple species. Is this referring to only microsporidia that don’t infect nematodes?

296 variable instead of flexible

299 How were spore clusters measured? What is their biological significance?

299: variation in polar tube length, rather than flexibility

306 “Together, our results support evolutionary change in both morphological features and specificity in Nematocida, notably with loss and gain of tissue specificity”

315-317: can the authors provide an analysis to support this claim? How conserved are metabolic proteins vs. host-interacting proteins?

333 do you mean: “...horizontal transfer of genes between microsporidia species…?”

338 “Microsporidia have been shown to display substantial variation in their phenotypic traits.” (consistent with our prior comments, flexibility and plasticity imply something more than what we think you mean here)

471-472 can the authors clarify this sentence “Families not classified as…” – what analysis was being done here?

497-500 indicate that a wide variety of doses were used across species in the exposure experiments. Is there a rationale for this and how does it affect interpretation of the results?

649, 664 presence

Reviewer #3: - The authors write that the nematode-infecting microsporidia Enterospectra and Pancytospora form a sister clade to Enterocytozoon (e.g. on line 92), and that Enterospectra and Pancytospora with Vittaforma corneae form a sister clade to Enterocytozoon, which appears more correct in their analysis (e.g. on lines 136, 160). Such assertion is contingent on the species they use for their analysis, so please make this clear that it is using the set of species shown in Fig. 1.

- Horizontal gene transfer of LGF1 family. In Fig. 4, only nematode-infecting microsporidia plus Vittaforma cornea are represented. Are all of these gene families absent in the other genomes shown in Fig 1?

Please explain how the horizontal transfer is inferred and explicit the scenarios in the section in lines 198-201.

line 200: the support on the tree in Fig S12 is not readable.

- Please indicate the reference for the raw sequence reads in Table S1. The accession number in the Methods (SRX13790799) seems to point only to reads to a Nematocida major archive.

lines 17-18: this sentence is not informative: basically any phylogenetic analysis would provide two groups of species. The same applies in other places in the manuscxript.

33: give an at least approximate genome size here.

58-60: within this general paragraph, do you now focus on nematode-infecting microsporidia in this sentence?

70-75: the succession of sentences is difficult to follow.

87: 'analysis'

93: sentence structure to revise, for example add 'that' before 'most'

104-107, etc: the logical flow is also difficult to follow here.

155: reveAled

184: such AS NemLGF11

188-90: convoluted sentence.

194: remove one 'this'

276, 279: do not use the word 'correlate' with a single example.

286: perhaps replace 'arranged' by 'plotted'.

297: ambiguity that could be resolved by 'both Nematocida groups containing species...'

307: 'throughout speciation' is unclear.

346: a stronger caveat is that this is not strongly supported, even using parsimony.

426, 431: explain why your have to remove ERTm1 contamination in the read data.

593: the GPS coordinates seem excessively precise to the nanometer for a millimeter long nematode.

641: Fig 1 legend: explain what the phylogenetic tree is based on.

649, 664: presence (typo)

672: change the title: Vittaforma cornea is not a nematode-infecting species.

698: typo PanAgrellus

705: 'a Nematocida'

926: rephrase as the L1 larvae do not stay at this stage for 96 hrs.

The respective predicted genome size of the nematode-infecting microsporidia is hard to read on Fig. 1 because of the large variation among all represented microsporidia and is only noted in Fig. 1. Figure S3 could have the species names next to each dot. And/or the genome sizes could be provided in Fig. 7.

Could you explain what Panagrellus "sp. 2" refers to?

Figure 1: make font sizes larger for the x axis and the legend.

Figure 3: is this figure important?

Figure 5: typo at 'epiphaga'

Figure 6C: explain what is to be seen in panel C. For example, for the species on the left, does it infect the intestine (this seems contradictory to panel D, is it a rare event?)? In the muscle panels, the staining of microsporidia does not seem to match that of the muscles.

PLOS authors have the option to publish the peer review history of their article (what does this mean?). If published, this will include your full peer review and any attached files.

Reviewer #1: No

Reviewer #2: No

Reviewer #3: No
---

## [Decision Letter · Decision Letter 1]

23 May 2023

Dear %TITLE% Reinke,

Thank you very much for submitting your manuscript "Genomic and phenotypic evolution of nematode-infecting microsporidia" for consideration at PLOS Pathogens. As with all papers reviewed by the journal, your manuscript was reviewed by members of the editorial board and by several independent reviewers. The reviewers appreciated the attention to an important topic. Based on the reviews, we are likely to accept this manuscript for publication, providing that you modify the manuscript according to the review recommendations.

The revised manuscript is much improved, and more coherently integrates the datasets. The added alternatives to the HGT hypothesis are welcomed, which was considered the major flaw in the prior version of the manuscript. One of the reviewers felt that some further clarification is necessary, and they provide detailed suggestions how best to address their reservations in text. In conclusion, the authors have generated a broad analysis of microsporidia paired with phenotyping of multiple closely-related and experimentally tractable species infecting nematodes. This paper is poised to become a valuable resource for the community and a source of interesting hypotheses for the field. Please specifically address the concerns raised by Reviewer 2 in the next iteration.

Sincerely,

Michael E. Grigg

Guest Editor

PLOS Pathogens

P'ng Loke

Section Editor

PLOS Pathogens

Kasturi Haldar

Editor-in-Chief

PLOS Pathogens

orcid.org/0000-0001-5065-158X

Michael Malim

Editor-in-Chief

PLOS Pathogens

orcid.org/0000-0002-7699-2064

The revised manuscript is much improved, and more coherently integrates the datasets. The added alternatives to the HGT hypothesis are welcomed, which was considered the major flaw in the prior version of the manuscript. One of the reviewers felt that some further clarification is necessary, and they provide detailed suggestions how best to address their reservations in text. In conclusion, the authors have generated a broad analysis of microsporidia paired with phenotyping of multiple closely-related and experimentally tractable species infecting nematodes. This paper is poised to become a valuable resource for the community and a source of interesting hypotheses for the field. Please specifically address the concerns raised by Reviewer 2 in the next iteration.

Reviewer Comments (if any, and for reference):

Reviewer's Responses to Questions

**Part I - Summary**

Reviewer #1: (No Response)

Reviewer #2: I re-reviewed this manuscript with a member of my lab group. The manuscript serves as a resource for the microsporidia and a source of hypotheses for the field. The presentation of the manuscript is improved, as is the presentation of the HGT hypothesis. However, we continue to have reservations about the presentation of this idea and the lack of supporting data. We detail these below.

**Part II – Major Issues: Key Experiments Required for Acceptance**

Reviewer #1: (No Response)

Reviewer #2: We appreciate the authors’ inclusion of a nuanced discussion of the different hypotheses to explain the presence of the NemLGF1 family in two distantly related genera. A few further modifications:

1.There are sentences in the abstract (line 41) and results (line 230) that present the HGT hypothesis as the explanation for this result. We agree that HGT is a viable hypothesis to explain the observed pattern. However, the manuscript does not include data to support this hypothesis. Speculation about the HGT hypothesis should therefore be restricted to the Discussion section.

2.Including the sequence identities is a good idea (lines 239-242), but in the absence of comparative data, these are not useful as data to support (or not) the HGT hypothesis. We need to know the expected sequence identity for proteins drawn at random from Nematocida and Pancytospora. As Reviewer 1 suggests, if the sequence identity for NemLGF1 members is higher than we’d expect given the genome wide averages, that would provide at least some quantitative support for the HGT hypothesis.

3.In lines 239-242 and 363-364, the authors make the point that the high structural similarity of the NemLGF1 proteins in spite of low sequence identities argues for HGT and against convergence. Frankly, this pattern sounds a bit like convergence – more similarity of properties related to function than you might expect given low underlying relatedness. We do agree with the authors’ intuition that convergence is an unlikely explanation, but we felt this claim about structural similarity did not help the case.

4.Following on this point, the authors argue that the NemLGF1 protein structures are highly similar, but we found this difficult to see in Fig. 5 – there’s some bits that seem to overlap, some that don’t. Is there a statistical framework in which to evaluate structural similarity of these predicted structures? This would help make the point. Moreover, we would benefit from an explanation in the text of how to interpret RSDM values for this purpose and what these values signify in the context of structural similarity.

**Part III – Minor Issues: Editorial and Data Presentation Modifications**

Reviewer #1: (No Response)

Reviewer #2: 89-91 revise for parallelism

100-102 with new additions to this paragraph, this sentence can be cut

378 “than” in place of “that”

400 perturbing

470 The authors first mention ERTM1 here when talking about removing contaminants. However, ERTM1 is not described as N. parisii until line 614.

489 Are the methods to predict proteins for M. incurvata and E. canceri the same as in 478?

515-516 does this mean that, in Fig. 4, families were marked as green or yellow only if >50% of proteins had a signal peptide or transmembrane domains? This point is critical to interpretation and wasn’t clear to us initially. Given that the methods come after the results, this methodological detail should be included in the results and figure 4 legend

Figure S1: can the authors add a matrix presenting similarity across species as well as this between-species matrix?

PLOS authors have the option to publish the peer review history of their article (what does this mean?). If published, this will include your full peer review and any attached files.

Reviewer #1: No

Reviewer #2: No

Figure Files:

Data Requirements:

Reproducibility:

References:

---

## [Editor Report · Decision Letter 2]

24 Jun 2023

Dear %TITLE% Reinke,

We are pleased to inform you that your manuscript 'Genomic and phenotypic evolution of nematode-infecting microsporidia' has been provisionally accepted for publication in PLOS Pathogens.

Best regards,

Michael E. Grigg

Guest Editor

PLOS Pathogens

P'ng Loke

Section Editor

PLOS Pathogens

Kasturi Haldar

Editor-in-Chief

PLOS Pathogens

orcid.org/0000-0001-5065-158X

Michael Malim

Editor-in-Chief

PLOS Pathogens

orcid.org/0000-0002-7699-2064

Reviewer Comments (if any, and for reference):

The authors have addressed the major concerns of all the reviewers, and it is now acceptable for publication

---

## [Editor Report · Acceptance letter]

13 Jul 2023

Dear Reinke,

We are delighted to inform you that your manuscript, "Genomic and phenotypic evolution of nematode-infecting microsporidia," has been formally accepted for publication in PLOS Pathogens.

Best regards,

Kasturi Haldar

Editor-in-Chief

PLOS Pathogens

orcid.org/0000-0001-5065-158X

Michael Malim

Editor-in-Chief

PLOS Pathogens

orcid.org/0000-0002-7699-2064